

# Dissolved carbon biogeochemistry and export in mangrove-dominated rivers of the Florida Everglades

David T. Ho[1], Sara Ferrón[1], Victor C. Engel[2,3], William T. Anderson[4,5], Peter K. Swart[6], René M. Price[4,5], Leticia Barbero[7]

5   [1]Department of Oceanography, University of Hawaii, Honolulu, Hawaii 96822, USA
[2]South Florida Natural Resources Center, Everglades National Park, Homestead, Florida 33030, USA
[3]Now at: U.S. Geological Survey, Wetland and Aquatic Research Center, Gainesville, Florida 32653, USA
[4]Southeast Environmental Research Center, Florida International University, Miami, Florida 33199, USA
[5]Department of Earth and Environment, Florida International University, Miami, Florida 33199, USA
10  [6]Marine Geosciences, Rosenstiel School of Marine and Atmospheric Science, University of Miami, Miami, Florida 33149, USA
[7]NOAA Atlantic Oceanographic and Meteorological Laboratory, Miami, Florida 33149, USA

*Correspondence to*: David T. Ho (david.ho@hawaii.edu)



**Abstract.** The Shark and Harney Rivers, located on the southwest coast of Florida, USA, originate in the freshwater, karstic marshes of the Everglades and flow through the largest contiguous mangrove forest in North America. In November 2010 and 2011, dissolved carbon source-sink dynamics were examined in these rivers during $SF_6$ tracer release experiments. Approximately 80% of the total dissolved carbon flux from all sources (i.e., freshwater wetlands, mangrove, carbonate dissolution, and marine input) out of the Shark and Harney Rivers during these experiments was as inorganic carbon, either via air-water $CO_2$ exchange or longitudinal flux of inorganic carbon to the coastal ocean. Of the total mangrove-derived dissolved inorganic carbon (DIC) exported from the forests into these rivers, between 42 and 48% was emitted to the atmosphere, with the remaining discharged to the coastal ocean. Dissolved organic carbon (DOC) represented ca. 10% of the total mangrove-derived dissolved carbon export from the forests. The sum of mangrove-derived DIC and DOC export to these rivers was estimated to be at least 18.9 to 24.5 mmol $m^{-2}$ $d^{-1}$, a rate lower than other independent estimates from Shark River and from other mangrove forests. Results from these experiments also suggest that in this region, mangrove contribution to the estuarine flux of dissolved carbon to the ocean is less than 10%.



## 1    Introduction

In many tropical and sub-tropical regions, mangrove forests are a typical feature surrounding estuaries (Twilley et al., 1992; Bouillon et al., 2008a). Mangroves are thought to play an important role in tropical and subtropical coastal biogeochemical cycling and the global coastal carbon budget, due to their high productivity and rapid cycling of organic and inorganic carbon (Twilley et al., 1992; Jennerjahn and Ittekkot, 2002; Dittmar et al., 2006). However, there remain uncertainties regarding the fate of mangrove-fixed carbon and the amount of carbon exported to the coastal waters from these ecosystems (Bouillon et al., 2008a; Bouillon et al., 2008b; Kristensen et al., 2008).

Bouillon et al. (2008a) showed that over 50% of the carbon fixed by mangroves through photosynthesis could not be accounted for by growth in biomass, accumulation in soils, and export of organic carbon, and    suggested that a large fraction of this missing organic carbon may be mineralized to dissolved inorganic carbon (DIC) and either lost to the atmosphere or exported to the  surrounding  waters. In fact, several studies have shown that the lateral advective transport of interstitial waters through tidal pumping represents a major carbon export pathway from mangroves into adjacent waters, both for DIC (Koné and Borges, 2008; Miyajima et al., 2009; Maher et al., 2013) and dissolved organic carbon (DOC) (Dittmar and Lara, 2001; Bouillon et al., 2007c). However, to date, lateral mangrove derived aquatic carbon fluxes (as a proportion of overall forest carbon mass balance) have only been estimated for short time periods and over limited spatial (e.g., plot) scales (e.g., Troxler et al., 2015). These studies also typically do not determine the fate of mangrove-derived carbon once it is exported from the forest through tidal pumping and drainage. Additional measurements of the magnitude and fate of mangrove carbon export at the basin scale are needed to help quantify connections between inter-tidal, estuarine and coastal ocean carbon cycles.

Basin-scale rates of lateral advective dissolved carbon export from a tidal mangrove forest are difficult to measure, because these fluxes are heterogeneous over space and time due to large variability in inundation patterns, forest structure, topography, and soil hydraulic properties. However, mangrove-derived dissolved carbon fluxes may be estimated in some systems using information on the spatial distribution of carbon-related measurements in adjacent waters. For example, the carbon balance of tidal riverine systems adjacent to mangrove forests should integrate the spatial and temporal variability of these lateral fluxes.



The objective in the study is to quantify dissolved carbon source-sink dynamics in a subtropical estuary

dominated by two tidal rivers, the Shark and Harney Rivers in Everglades National Park, Florida, USA. These rivers

are centrally-located within the largest contiguous mangrove forest in North America and they discharge to the Gulf

of Mexico. The total dissolved carbon inventories and fluxes in these rivers are determined using a series of discrete

and continuous measurements of carbon-related parameters along a salinity gradient, and the mangrove contribution

separated using measurements of stable isotopic composition of dissolved organic and inorganic carbon. The results

are then scaled by the area of mangrove forest that surrounds these rivers to express dissolved carbon fluxes on an

aerial basis for comparison to independent measurements of dissolved carbon fluxes from this forest.

## 2    Methods

### 2.1    Study site

The tidal-dominated Shark and Harney Rivers (river and estuary are used interchangeably in this

contribution) are surrounded by mangrove forests and located on the southwest coast of Florida (Fig. 1), within

Everglades National Park. The subtropical climate in southern Florida is characterized by a May to October wet

season, when approximately 60% of the annual precipitation occurs (Southeast Regional Climate Center,

http://www.sercc.com). The Shark and Harney Rivers together discharge approximately 50% of the flow from the

Shark River Slough (SRS), the primary drainage feature of Everglades National Park, to the Gulf of Mexico (GOM)

(Levesque, 2004). Seasonal variation of the water discharge from SRS mostly follows the precipitation patterns

(Saha et al., 2012), and influences the transport of nutrients to the mangrove ecotone (Rivera-Monroy et al., 2011).

The Shark and Harney Rivers are each approximately 15 km long, and connected in Tarpon Bay (Fig. 1). The mean

depths of Tarpon Bay, Shark River, and Harney River at mid tide are $1.4 \pm 0.3$, $2.8 \pm 0.4$, and $2.6 \pm 0.4$ m (Ho et al.,

2014), respectively, and the surface areas are $1.48 \times 10^6$, $2.54 \times 10^6$, and $2.75 \times 10^6$ m$^2$, respectively. The inter-tidal

zones bordering the Shark and Harney Rivers are dominated by *Rhizophora mangle* (red mangrove), *Avicennia*

*germinans* (black mangrove), *Laguncularia racemose* (white mangrove), and *Conocarpus erectus* (buttonwood).

Semi-diurnal tides in this region inundate the forest as often as twice a day. River discharge to the GOM is primarily

influenced by tides, wind, and freshwater inflow from SRS (Levesque, 2004).

Discharges are determined by the US Geological Survey at stations near the midpoints of Shark River

(USGS 252230081021300 Shark River) and Harney River (USGS 252551081050900 Harney River) (Fig. 1).



Discharges are generally lower during March-May than the rest of the year. Hourly mean residual discharge values (i.e., filtered for tides) from March to May of the 5-year period from 2007 to 2011 ranged from -21.9 to 24.1 $m^3 s^{-1}$, with a mean of 0 $m^3 s^{-1}$ for Shark River, and ranged from -28.9 to 38.5 $m^3 s^{-1}$, with a mean of 4.4 $m^3 s^{-1}$ for Harney

River. Positive values indicate flow towards the GOM. For the rest of the year (i.e., June to February), these values ranged from -46.2 to 89.2 $m^3 s^{-1}$, with a mean of 8.8 $m^3 s^{-1}$ for Shark River, and -41.6 to 75.0 $m^3 s^{-1}$, with a mean of 11.3 $m^3 s^{-1}$ for Harney River.

### 2.2 Shark River Tracer Release Experiments

Two field studies were conducted as part of the Shark River Tracer Release Experiment (SharkTREx 1: 20

to 25 November 2010; SharkTREx 2: 10 to 15 November 2011; (Ho et al., 2014)). The mean residual discharges for Shark River were 6.9 (hourly range: -2 to 19.9) and 4.9 (hourly range: -18.9 to 34.8) $m^3 s^{-1}$, during SharkTREx 1 and 2, respectively, and those for Harney River were 6.0 (hourly range: -1.6 to 22.8) and 1.9 (hourly range: -17.3 to 30.6) $m^3 s^{-1}$, during SharkTREx 1 and 2, respectively (U.S. Geological Survey, 2016).

During both campaigns, an inert tracer (sulfur hexafluoride; $SF_6$) was injected in the river near the point

where the rivers diverge just downstream of Tarpon Bay (25.4092, -81.0083) to determine the rates of longitudinal dispersion, and the water residence time. Each day, longitudinal surveys were made along the Shark and Harney Rivers from Tarpon Bay to the GOM, and included continuous underway measurements of temperature, salinity, $SF_6$, dissolved $O_2$ (DO; $\mu mol\ kg^{-1}$), and partial pressure of $CO_2$ (pCO$_2$; $\mu atm$), and discrete measurements of total alkalinity (TAlk; $\mu mol\ kg^{-1}$), dissolved inorganic carbon (DIC; $\mu mol\ kg^{-1}$), dissolved organic carbon (DOC; $\mu mol$

$kg^{-1}$), stable carbon isotopic composition of DIC and DOC ($\delta^{13}C_{DIC}$ and $\delta^{13}C_{DOC}$, respectively; ‰).

### 2.3 Discrete measurements

During SharkTREx 1, three to five surface water samples were collected daily in the Shark River with a 5-L Niskin bottle at ~0.5 m below the surface for the analysis of TAlk, DOC, $\delta^{13}C_{DIC}$, and $\delta^{13}C_{DOC}$. At each sampling site, vertical profiles of temperature, salinity, and DO were recorded using a conductivity, temperature, and depth

sonde (Sea-Bird SBE 19plus V2) equipped with a Clark type polarographic $O_2$ sensor (SBE 43). These profiles showed that the water column was vertically well mixed. No discrete samples were collected in the Harney River during SharkTREx 1. During SharkTREx 2, discrete samples for DIC, TAlk, DOC, $\delta^{13}C_{DIC}$, and $\delta^{13}C_{DOC}$ were collected daily at 20 stations distributed within the Shark and Harney Rivers (Fig. 1).



### 2.3.1 Total alkalinity and dissolved inorganic carbon

During SharkTREx 1, samples for TAlk were collected in 250 mL HDPE bottles after passing through a 0.45 μm filter. They were stored on ice for transport to the laboratory at FIU, where TAlk was determined at room temperature using an automated titrator (Brinkman Titrino 751) with 0.1 N HCl to a pH of 2. TAlk was calculated from the volume of acid added at the inflection point closest to a pH of 4, and reported as meq $L^{-1}$ $HCO_3^-$ since the original pH of the water samples was near neutral. The precision of the measurements was ±2% from replicate

analysis ($n$ = 5) with an accuracy of ±2% as determined by analysis of certified reference material (Dickson, 2010). DIC and pH were computed from TAlk and $pCO_2$ using the dissociation constants of (Millero, 2010) for estuarine waters.

      During SharkTREx 2, samples for TAlk and DIC were collected in 550 mL borosilicate glass bottles, poisoned with $HgCl_2$, and sealed with hydrocarbon grease (Apiezon M). The samples were stored at room

temperature in the dark for travel to the laboratory at NOAA/AOML. Samples for TAlk were measured in an open thermostated cell (25 °C) with an automated titrator (Metrohm 765 Dosimat) connected to a pH glass-reference electrode system (Orion), using 0.2 M HCl as a titrant, and determined from the equivalence point of the titration curve using a non-linear least-squares fit. For DIC analysis, water samples were first acidified to convert all the carbonate species to $CO_2$ in a DIC analyzer (Apollo SciTech), and then measured with a NDIR detector (LI-COR

LI-7000). Calibrations for DIC and TAlk were performed using certified reference material (Dickson, 2010). The analytical uncertainty of the DIC and TAlk measurements based on replicate samples are 0.1 and 0.2%, respectively.

      The measured TAlk and $pCO_2$ from SharkTREx 2 were used to calculate DIC using CO2SYS (Pierrot et al., 2006) and the dissociation constants of (Millero, 2010), and the results were 1.3 ± 1.1% (range: -2.4 to +4.4%) higher than the measured DIC, possibly indicating a slight contribution (ca. 1%) to TAlk from organic or particulate

material, as the samples were not filtered.

### 2.3.2 Dissolved organic carbon

      The samples analyzed for DOC were filtered with pre-combusted 0.7 μm GF/F filters and collected in pre-cleaned, acid-washed, brown high-density polyethylene bottles (HDPE; Nalgene). Containers were rinsed three times before sample collection, transported on ice to the FIU SERC Nutrient Analysis Lab, and stored in a

refrigerator until analyses within three weeks of collection. DOC was measured using the high-temperature catalytic combustion method on a total organic carbon analyzer (Shimadzu TOC-V).




### 2.3.3 Stable carbon isotopic composition

Samples for $\delta^{13}C_{DIC}$ were collected in 40 ml glass bottles after passing the sample through a GF/F filter, and then poisoning with $HgCl_2$. In the laboratory at RSMAS, vials with 0.5 ml 103% $H_3PO_4$ were flushed for 60 s

with He. Approximately 2 ml of sample were then injected into the vial, and after sonification the accumulated $CO_2$ was analyzed by a gas chromatograph (GC) coupled to an isotope ratio mass spectrometer (GC-IRMS; Thermo Delta V). The $\delta^{13}C$ was calibrated using two standards of $NHCO_3$ with differing $\delta^{13}C$ values dissolved in $H_2O$ whose isotopic compositions had been previously calibrated relative to NBS-19 using conventional dual inlet mass spectrometry (Finnigan-MAT 251). The $\delta^{13}C$ values are reported relative to the Vienna Pee Dee Belemnite (VPDB)

standard.

Samples for $\delta^{13}C_{DOC}$ were collected in 60 ml brown HDPE bottles and stored on ice until returned to the lab at FIU. $\delta^{13}C_{DOC}$ samples were filtered with GF/F (0.7 μm) filter, and then stored in pre-cleaned 40 ml bottles until analyses. Measurements for $\delta^{13}C_{DOC}$ were made using a total organic carbon (TOC) analyzer (Aurora 1030W, OI Analytical) coupled to a cavity ring-down spectroscopy system (CRDS; G1111-i, Picarro) following the approach of

Ya et al. (2015). DIC was removed by adding phosphoric acid and sparging with $N_2$ gas. 1.5 ml of sample was chemically oxidized to $CO_2$ at a temperature of 98°C in the presence of sodium persulfate ($Na_2S_2O_8$). The $CO_2$ generated was detected by non-dispersive infrared absorption (NDIR) for determination of DOC. The $CO_2$ was collected in a gas-tight bag and then pulsed into the CRDS for the $\delta^{13}C$ measurement. Analytical precision based on replicated standards ranged from ±0.15 to ±1.52 ‰ for this study.

### 2.4 Underway measurements

Surface water was continuously pumped from an intake located near the bow of the boat at a water depth of approximately 1 m during tracer recovery operations. Water temperature and salinity were continuously recorded using a thermosalinograph (SBE 45 MicroTSG). During SharkTREx 1, DO was measured underway with a membrane covered galvanic sensor (WTW Cellox 325) calibrated with saturated air. During SharkTREx 2, DO was

measured using an oxygen optode (Aanderaa 3835) calibrated against Winkler titration.

Underway measurements of atmospheric and waterside $pCO_2$ were made. Waterside $pCO_2$ were obtained with a showerhead type equilibrator coupled to a non-dispersive infrared (NDIR) analyzer (LI-COR 840A). Measurements of underway $SF_6$ were made with an automated $SF_6$ analysis system (Ho et al., 2002). Both of these measurements have been described in detail in Ho et al. (2014)

### 2.5 Inventories of DIC, DOC and DO

The inventories of DIC, DOC and DO were calculated in the same way that $SF_6$ inventories were determined in Ho et al. (2014). The river was divided into 100-m longitudinal sections, and the measured concentrations, corrected for tidal movement to slack before ebb for each day, were assigned to each section $i$ and



then summed over the entire length of the river. For example, to calculate the inventory of DIC, denoted

$\sum[\text{DIC}]_{observed}$ (mol):

$$\sum[\text{DIC}]_{observed} = \sum_{i=1}^{n}[\text{DIC}]_i \times V_i, \qquad\qquad (1)$$

where $[\text{DIC}]_i$ is the mean concentration (mol L$^{-1}$) in section $i$, $V_i$ is the volume of the river (L) in section $i$ at mid-

tide, and $n$ is the number of sections in each river ($n$ = 273 for Shark River and Tarpon Bay; $n$ = 152 for Harney

River). DOC and DO inventories were also calculated using Eq. (1), by substituting $[\text{DOC}]_i$ or $[\text{DO}]_i$ for $[\text{DIC}]_i$

accordingly. The inventories of DIC and DOC were separated into contributions from estuarine and non-estuarine

sources, first by determining inventories for DIC assuming conservative mixing between the freshwater and marine

end members and then subtracting these inventories from the total observed inventories.     The estuarine DIC

inventory, $\sum[\text{DIC}]_{estuary}$, representing the DIC from all estuarine sources, was calculated as follows:

$$\sum[\text{DIC}]_{estuary} = \sum[\text{DIC}]_{observed} - \sum[\text{DIC}]_{conserv} + \sum[\text{DIC}]_{gasex}, \qquad (2)$$

where $\sum[\text{DIC}]_{conserv}$ is the inventory of DIC assuming conservative mixing between freshwater and marine end

members (i.e., from non-estuarine sources), and $\sum[\text{DIC}]_{gasex}$ is the inventory of DIC lost to air-water gas exchange

from the estuary, due to $pCO_2$ in the water being above solubility equilibrium with the atmosphere (see section 2.6).

The freshwater and marine end-members were assigned to the values measured at the lowest (Tarpon Bay) and

highest salinities, respectively.

The total $O_2$ deficit in Shark River during the experiments was determined by examining the difference in

$O_2$ inventories for conservative mixing and actual measurements, correcting for $O_2$ influx due to gas exchange using

a formulation similar to Eq. (2) above (i.e., $\sum[\text{DO}]_{deficit} = \sum[\text{DO}]_{conserv} - \sum[\text{DO}]_{observed} + \sum[\text{DO}]_{gasex}$).

### 2.6     Air-water $O_2$ and $CO_2$ fluxes

To enable comparison between different gases and different aquatic environments, it is customary to

normalize gas transfer velocities to a Schmidt number ($Sc$; kinematic viscosity of water divided by diffusion

coefficient of gas in water) of 600, $k(600)$, corresponding to that of $CO_2$ in freshwater at 20 °C. $k(600)$ for

SharkTREx 1 and 2, determined from the parameterization proposed in Ho et al. (2016), were 3.5 ± 1.0 and 4.2 ±

1.8 cm h$^{-1}$, respectively. To determine $k$ for $O_2$ and $CO_2$ at the temperature and salinity measured in the rivers, the

following equation was used, assuming a $Sc^{-1/2}$ scaling (Jähne et al., 1987):

$$k_{O_2} = k(600)\left(\frac{Sc_{O_2}}{600}\right)^{-1/2}, \qquad\qquad (3)$$



where $k$ and $Sc$ of $CO_2$ could be substituted in Eq. (3) for $O_2$, and $Sc$ for $O_2$ and $CO_2$ were calculated as a function of

temperature and salinity using data compiled by Wanninkhof (2014).

Air-water $O_2$ fluxes ($F_{O_2}$; mmol m$^{-2}$ d$^{-1}$) were calculated as follows:

$$F_{O_2} = k_{O_2} \left( O_{2_{equil}} - O_2 \right), \tag{4}$$

where $k_{O_2}$ (cm h$^{-1}$) is the gas transfer velocity for $O_2$, $O_{2_{equil}}$ (mmol m$^{-3}$) is the equilibrium concentration of $O_2$ in

the water at a given temperature and salinity (Garcia and Gordon, 1992), and $O_2$ is the measured oxygen

concentration in the water.

Similarly, air-water $CO_2$ fluxes ($F_{CO_2}$; mmol m$^{-2}$ d$^{-1}$), which were used to determine changes in DIC due to

gas exchange, were calculated as follows:

$$F_{CO_2} = k_{CO_2} K_0 \Delta pCO_2, \tag{5}$$

where $k_{CO_2}$ (cm h$^{-1}$) is the gas transfer velocity for $CO_2$, $K_0$ (mol atm$^{-1}$ m$^{-3}$) is the aqueous-phase solubility of $CO_2$

(Weiss, 1974), and $\Delta pCO_2$ (µatm) is the difference between the measured $pCO_2$ in air equilibrated with water and

atmospheric $pCO_2$.

As with the inventories, $F_{CO_2}$ were separated into estuarine and non-estuarine contributions. Because of the

non-linearity in the relationship between $pCO_2$ and other carbonate system parameters, the $pCO_2$ in the river

expected from conservative mixing was calculated by assuming conservative mixing for DIC and TAlk, and then

calculating $pCO_2$ using CO2SYS (Pierrot et al., 2006), with the dissociation constants of Millero (2010). Then, the

non-estuarine $F_{CO_2}$ was calculated as above with Eq. (5), and the $F_{CO_2}$ attributed to estuarine sources was determined

as the difference between total and non-estuarine $F_{CO_2}$.

**2.7   Estuarine and mangrove contributions to DIC**

 DIC in the Shark and Harney Rivers may originate from several sources in addition to input from the freshwater

marsh upstream and the coastal ocean, including: 1) mangrove root respiration; 2) organic matter mineralization in

sediments or in river water; 3) dissolution of $CaCO_3$ in sediments or in river water; and 4) groundwater discharge.

Groundwater in this region is likely to contain DIC from $CaCO_3$ dissolution that occurs when saltwater intrudes into

the karst aquifer that underlies this region (Price et al., 2006). In this setting, the combination of #1 and #2

represents the mangrove source of DIC ([DIC]$_{mangrove}$), and the combination of #3 and #4 represents the $CaCO_3$

dissolution source ([DIC]$_{dissolution}$) to estuarine [DIC]:



$$[DIC]_{estuary} = [DIC]_{observed} - [DIC]_{conserv} + [DIC]_{gasex} = [DIC]_{mangrove} + [DIC]_{dissolution} \quad (6)$$

where $[DIC]_{observed}$ is the observed DIC concentration, $[DIC]_{conserv}$ is the DIC concentration expected by conservative

mixing of the two end-members, and $[DIC]_{gasex}$ is the correction for change in $[DIC]_{observed}$ due to loss through air-

water gas exchange as the water transits through the estuary. $[DIC]_{gasex}$ was determined from $F_{CO_2}$ and the residence

time of water during each experiment (Ho et al., 2016).

Measurements of $\delta^{13}C_{DIC}$ and estuarine DIC/TAlk ratios were used to determine the mangrove sources to

estuarine DIC. Fixation of $CO_2$ through photosynthesis is neglected in both models as these rivers are characterized

by low chlorophyll-*a* concentration and low phytoplankton biomass (Boyer et al., 1997). During SharkTREx 1 and

2, there was a negligible difference between $pCO_2$ measured during the day and night.

### 2.7.1    Determining mangrove contribution from $\delta^{13}C_{DIC}$

Processes 1 through 4 listed above influence $\delta^{13}C_{DIC}$ in the estuary differently due to the differences in the

$\delta^{13}C$ values originating from respiration of mangrove-derived organic matter, and $CaCO_3$ dissolution. The isotopic

fractionation during respiration of organic matter is small, and the $\delta^{13}C_{DIC}$ values produced via this pathway should

be equivalent to the $\delta^{13}C$ of the organic matter respired (DeNiro and Epstein, 1978). Also, the isotopic fractionation

during dissolution/re-precipitation of $CaCO_3$ is thought to be negligible (Salomons and Mook, 1986).

The expected $\delta^{13}C$ values of DIC in the rivers as a result of conservative mixing ($\delta^{13}C_{conserv}$) of the marine

and freshwater end-members of the Shark and Harney Rivers were calculated as follows (Mook and Tan, 1991):

$$\delta^{13}C_{conserv} = \frac{S([DIC]_F\delta^{13}C_F - [DIC]_M\delta^{13}C_M) + S_F[DIC]_M\delta^{13}C_M - S_M[DIC]_F\delta^{13}C_F}{S([DIC]_F - [DIC]_M) + S_F[DIC]_M - S_M[DIC]_F}, \quad (7)$$

where $[DIC]$ is the observed DIC concentration, $S$ is the measured salinity, and $M$ and $F$ subscripts refer to the

marine and freshwater end-members, respectively.

An estimate of the maximum contribution of $[DIC]_{mangrove}$ and $[DIC]_{dissolution}$ to $[DIC]_{estuary}$ can be obtained

by solving Equations 6 and 8:

$$\delta^{13}C_{DIC} \times [DIC]_{observed} = \delta^{13}C_{conserv} \times [DIC]_{conserv} + \delta^{13}C_{mangrove} \times [DIC]_{mangrove} + \delta^{13}C_{dissolution} \times$$

$$[DIC]_{dissolution} - \left(\delta^{13}C_{DIC} - \varepsilon_{^{13}C_{DIC-CO_2}}\right) \times [DIC]_{gasex}, \quad (8)$$

where the $\delta^{13}C_{conserv}$ value is the DIC isotopic composition expected for conservative mixing (Mook and Tan, 1991),

$\delta^{13}C_{mangrove}$ is the isotopic composition for mangrove-derived material (-30‰; Mancera-Pineda et al., 2009), the



$\delta^{13}C_{dissolution}$ value is the $\delta^{13}C$ composition of calcite (~1‰), and $\epsilon_{13C_{DIC-CO_2}}$ is the equilibrium isotope fractionation

between DIC and $CO_2$ gas (~8‰; Zhang et al., 1995).

### 2.7.2    Determining mangrove contribution from TAlk/DIC

An independent approach to separate the mangrove contribution from $CaCO_3$ dissolution is to use the co-variation of $[DIC]_{estuary}$ and $[TAlk]_{estuary}$ as an indicator of the biogeochemical processes affecting DIC dynamics (Borges et al., 2003; Bouillon et al., 2007c), as these processes have different effects on DIC and TAlk. Assuming

that $[TAlk]_{estuary}$ is mainly produced by the dissolution of $CaCO_3$, $[DIC]_{dissolution}$ can be determined as 0.5 x $[TAlk]_{estuary}$, and then $[DIC]_{mangrove}$ can be calculated from Eq. (6). However, since sulfate reduction, a primary mineralization pathway in mangrove sediments, may also contribute to [TAlk] (Alongi, 1998; Alongi et al., 2005) this calculation represents an upper bound estimate for $[DIC]_{dissolution}$ and a lower bound estimate for $[DIC]_{mangrove}$.

### 2.8    Determining mangrove contribution to DOC

In the Shark and Harney Rivers, dissolved organic matter may be derived from upstream freshwater wetland species such as periphyton and sawgrass, from seagrass communities and marine phytoplankton, or from mangrove vegetation inside the estuary (Jaffe et al., 2001). The estuarine contributions to DOC ($[DOC]_{estuary}$) in the rivers was determined in the same way as for DIC above using Eq. (6), by substituting DOC for DIC accordingly, without the correction for gas exchange:

$$[DOC]_{estuary} = [DOC]_{observed} - [DOC]_{conserv}, \qquad (9)$$

where $[DOC]_{observed}$ is the observed DOC concentration, $[DOC]_{conserv}$ is the DOC concentration expected from conservative mixing of the two end-members.

Then, measurements of $\delta^{13}C_{DOC}$ were made to ascertain the mangrove source of DOC in the river, in order to determine the proportion of $[DOC]_{estuary}$ that is of mangrove origin. The expected $\delta^{13}C$ values of DOC as a result

of conservative mixing ($\delta^{13}C_{conserv}$) were calculated using Eq. (7), substituting DOC for DIC. Assuming that $[DOC]_{estuary}$ was entirely mangrove-derived, $[DOC]_{mangrove}$ should equal:

$$[DOC]_{mangrove} = \frac{[DOC]_{observed}\delta^{13}C_{DOC} + [DOC]_{conserv}\delta^{13}C_{conserv}}{\delta^{13}C_{mangrove}}, \qquad (10)$$

where $\delta^{13}C_{mangrove}$ is the isotopic composition for mangrove-derived material (-30‰).





### 2.9 Longitudinal dispersion

The longitudinal $SF_6$ distribution was corrected for tidal movement to slack water before ebb for each day

using a method described in Ho et al. (2002). The absolute magnitudes of the average daily corrections were 2.0 and

2.7 km for SharkTREx 1 and 2, respectively, with a range for individual measurements of 0 to 5.8 km and 0 to 7.3

km for SharkTREx 1 and 2, respectively. Longitudinal dispersion coefficient $K_x$ (m$^2$ s$^{-1}$) was calculated from the

change of moment of the longitudinal $SF_6$ distribution over time as follows (Fischer et al., 1979; Rutherford, 1994):

$$K_x = \frac{1}{2}\left(\frac{d\sigma_x^2}{dt}\right),$$     (11)

where $\sigma_x^2$ is the second moment of the longitudinal $SF_6$ distribution for each day.

### 2.10 Longitudinal fluxes to the Gulf of Mexico

        The longitudinal fluxes of DIC and DOC from Shark and Harney Rivers to the Gulf of Mexico were

calculated using the averaged DIC or DOC inventories, and the residence time of water ($\tau$; d), which was determined

from the decrease in the inventory of $SF_6$ after correcting for air-water gas exchange (Ho et al., 2016). For example,

the longitudinal DIC flux ($F_{DIC}$; mol d$^{-1}$) can be calculated as follows:

$$F_{DIC} = \frac{\sum[DIC]_{observed}}{\tau}.$$     (12)

Equation (12) can be used to calculate the fluxes of any other dissolved or suspended substance in the river by

substituting its inventory in place of DIC. As with the inventory calculations, longitudinal fluxes were separated into

estuarine and non-estuarine contributions.

        The advantage of this method to calculate longitudinal flux in a tidal river over a method that uses net

discharge and constituent concentration is that the effect of tidal flushing is implicitly accounted for by the residence

time, and therefore there is not a need to explicitly define the fraction of river water in the return flow during each

flood tide.

## 3 Results and Discussion

### 3.1 Distribution patterns and carbon inventories

        During SharkTREx 1, the salinity along the longitudinal transects ranged from 1.2 to 27.1, and the mean ($\pm$

s.d.) water temperature was 23.4 $\pm$ 0.2 °C. During SharkTREx 2, salinity ranged from 0.6 to 27.1, and water

temperatures averaged 22.7 $\pm$ 0.9 °C.

Both $pCO_2$ and DO showed large spatial variability within the Shark and Harney Rivers during SharkTREx

1 and 2 (Fig. 2). Measured $pCO_2$ values were well above atmospheric equilibrium along the entire salinity range,



with values ranging from ca. 1000 to 6200 µatm. Maximum $pCO_2$ values were observed at intermediate salinities, decreasing towards both end-members, while DO showed the opposite pattern, with saturations ranging from 36 to 113%.

The patterns of TAlk and DIC along the salinity gradient followed the same trend as $pCO_2$ and were clearly non-conservative (Fig. 3a-f). TAlk varied between ca. 3400 and 5000 µmol kg$^{-1}$ during SharkTREx 1 and between ca. 3000 and 3900 µmol kg$^{-1}$ during SharkTREx 2. DIC ranged from ca. 3400 to 5100 µmol kg$^{-1}$ during SharkTREx 1, and ca. 2800 to 4000 µmol kg$^{-1}$ during SharkTREx 2. $\delta^{13}C_{DIC}$ values ranged from -10.3 to -6.6 ‰ and from -11.4 to -5.8 ‰ during SharkTREx 1 and 2, respectively. Higher DIC, TAlk and $pCO_2$ coincided with lower $O_2$ saturation,

more depleted $\delta^{13}C_{DIC}$, and lower pH values (Fig. 3g-i), indicative of mineralization of mangrove-derived organic matter within the estuary. A negative correlation was observed between $\delta^{13}C_{DIC}$ and both DIC and $pCO_2$ values, demonstrating that the source of estuarine DIC was depleted in $^{13}C$.

During SharkTREx 1, the DOC concentrations in the freshwater end member were higher than SharkTREx 2 (Fig. 4). For both experiments, DOC concentrations followed a non-conservative pattern (see also Cawley et al.,

2013), but this trend was less apparent during SharkTREx 1 compared to SharkTREx 2 (Fig. 4).

The inventories of DIC, DOC, DO, TAlk, and $pCO_2$ were relatively constant in the Shark and Harney Rivers, indicating quasi steady state conditions during SharkTREx 1 and 2. Under these conditions, carbon inputs and exports are balanced, and fluxes and concentrations may be examined interchangeably. $K_x$ during the experiments (16.4 ± 4.7 and 77.3 ± 6.5 m$^2$ s$^{-1}$ for Shark River during SharkTREx 1 and 2, respectively, and 136.1 ±

16.5 m$^2$ s$^{-1}$ for Harney River during SharkTREx 2) were relatively large, and suggest that any perturbations (such as export of DIC from mangroves) would be quickly mixed thoroughly in the estuary.

In the following, for brevity, fluxes and inventories are summarized as ranges, which cover the two rivers and two experiments so they reflect both temporal and spatial variability. The individual values are given in Tables 1 and 2.

DIC was the dominant form of dissolved carbon in both rivers and accounted for 79 to 82% of the total dissolved carbon in the rivers. The contribution of DOC to the total carbon pool varied between 18 and 21% (Table 1).




### 3.2 Air-water CO$_2$ fluxes

As shown by Ho et al. (2014), pCO$_2$ observed during SharkTREx 1 and 2 fall in the upper range of those

reported in other estuarine (Borges, 2005) and mangrove-dominated systems (Bouillon et al., 2003; Bouillon et al.,

2007a; Bouillon et al., 2007b; Koné and Borges, 2008; Call et al., 2015). The mean air-water CO$_2$ fluxes in Shark

River for SharkTREx 1 and 2 were $105 \pm 9$ and $99 \pm 6$ mmol m$^{-2}$ d$^{-1}$, (Ho et al., 2016). The analysis is taken further

here by including data from Harney River. The mean air-water CO$_2$ fluxes in Harney River were $150 \pm 8$ and $114 \pm$

$21$ mmol m$^{-2}$ d$^{-1}$ for SharkTREx 1 and 2, respectively.

Borges et al. (2003) summarized all available pCO$_2$ data from mangrove surrounding waters, and calculated

CO$_2$ fluxes to the atmosphere that averaged 50 mmol m$^{-2}$ d$^{-1}$ (with a range of 4.6 to 113.5 mmol m$^{-2}$ d$^{-1}$), and

Bouillon et al. (2008a) estimated a global CO$_2$ flux from mangroves of ca. $60 \pm 45$ mmol m$^{-2}$ d$^{-1}$. One reason that the

fluxes from SharkTREx 1 and 2 are on the upper end of those estimates may be that the Shark and Harney Rivers

receive a large input of DIC from the freshwater marsh upstream (Table 1), causing higher pCO$_2$ in the estuary

compared to the global average.

Scaling the air-water CO$_2$ fluxes by the area of open water in the Shark and Harney Rivers, where Tarpon

Bay is included with Shark River, suggests that the total carbon emissions to the atmosphere through air-water gas

exchange in Shark River was $4.2 \pm 0.4 \times 10^5$ and $4.0 \pm 0.2 \times 10^5$ mol d$^{-1}$ during SharkTREx 1 and 2, respectively,

and were $4.1 \pm 0.2 \times 10^5$ and $3.1 \pm 0.6 \times 10^5$ mol d$^{-1}$ from the Harney River during SharkTREx 1 and 2, respectively

(Fig. 5), which is remarkably consistent, both spatially and temporally.

These fluxes were incorporated into the DIC mass balance of the Shark and Harney Rivers (Eq. 2) by

calculating the total CO$_2$ degassed over the residence time of water in the rivers. Given the mean air-water CO$_2$

fluxes (Table 2), the total CO$_2$ degassed in the Shark River represents approximately 13 and 21% of $\sum[DIC]_{observed}$

during SharkTREx 1 and 2, respectively, and the CO$_2$ degassed from the Harney River during SharkTREx 2

represents 20% of $\sum[DIC]_{observed}$, indicating that air-water CO$_2$ exchange removes a non-negligible fraction of the

inorganic carbon in these rivers. Exclusion of $\sum[DIC]_{gasex}$ from the mass balance in Eq. (2) would lead to an

underestimation of $\sum[DIC]_{estuary}$ of between 33 and 44%.





### 3.3 Mangrove contribution to DIC inventory

The highest DIC concentrations were correlated with low DO (Fig. 2) and characterized by [13]C-depletion

(Fig. 3j, k, l). Observations of elevated DIC and $pCO_2$ in the middle of the estuary, coupled with $\delta^{13}C_{DIC}$ and $O_2$

depletion may indicate the importance, noted by other authors, of lateral transport of pore water from the peat-based

mangrove forest into the river via tidal pumping (Bouillon et al., 2008a; Maher et al., 2013). However, as

demonstrated below, the observed DIC and $\delta^{13}C_{DIC}$ distributions in these rivers cannot be explained solely by

mineralization of mangrove-derived organic carbon.

**3.3.1 Evidence from $\delta^{13}C_{DIC}$**

The distributions of DIC and $\delta^{13}C_{DIC}$ cannot be explained solely by the addition of mangrove-derived DIC

and air-water gas exchange. Solving Eq. (8) for $\delta^{13}C_{DIC}$, assuming that $[DIC]_{dissolution}$ is negligible and that the only

source of DIC in the rivers is of mangrove origin, would result in $\delta^{13}C$ values significantly lower than those

observed. The low pH in interstitial waters of mangrove sediments due to organic matter mineralization processes

may be favorable to $CaCO_3$ dissolution in mangrove sediments, and this process could have an effect on estuarine

$\delta^{13}C_{DIC}$. Groundwater discharge could also influence DIC and $\delta^{13}C_{DIC}$. Inputs of DIC derived from $CaCO_3$

dissolution from either of these sources may explain the differences in observed $\delta^{13}C_{DIC}$ and those expected if

$[DIC]_{estuary}$ was entirely of mangrove origin.

Other recent studies have observed similar differences in pore water $\delta^{13}C_{DIC}$ values and the values expected

from the carbon inputs derived from organic matter decomposition and $CaCO_3$ dissolution (Walter et al., 2007). In

these studies, the differences were attributed to isotopic exchange during $CaCO_3$ dissolution and re-precipitation,

and if these processes were active in the substrate or mangrove sediments surrounding the Shark and Harney Rivers,

they would affect observed $\delta^{13}C_{DIC}$ values without affecting $[DIC]_{estuary}$.

Solving Equations 6 and 8, the mineralization of mangrove-derived organic matter is estimated to account

for ca. 60 ± 6 % of $\sum[DIC]_{estuary}$ (Table 3), with the remainder originating from the dissolution of $CaCO_3$. This

estimate is sensitive to the end member value chosen for $\delta^{13}C_{mangroves}$ and $\delta^{13}C_{dissolution}$. For instance, if $\delta^{13}C_{mangroves}$

were -29‰ instead of -30‰, the mangrove contribution would increase to 62%.



### 3.3.2 Evidence from DIC and TAlk

In the Shark and Harney Rivers, the high correlation ($r^2 = 0.99$; Fig. 6) between $[DIC]_{estuary}$ and $[TAlk]_{estuary}$

indicates the same processes control the inputs of DIC and TAlk to these rivers. By examining the covariation of

$[DIC]_{estuary}$ and $[TAlk]_{estuary}$, mangroves were found to contribute a minimum of $70 \pm 3$ % of $\sum[DIC]_{estuary}$ (Table

3), with the remainder due to the dissolution of $CaCO_3$. These estimates are in reasonable agreement with those

based on the carbon isotopic mass balance. One reason why the $\delta^{13}C_{DIC}$-based estimates might be lower is that the

potential of $^{13}C$ isotopic exchange during $CaCO_3$ dissolution and re-precipitation is not being considered.

The $[TAlk]_{estuary}$ vs. $[DIC]_{estuary}$ ratios were 0.84 and 0.92 for Shark River during SharkTREx 1 and 2, and

0.90 for the Harney River during SharkTREx 2 (Fig. 6). Given the estimated contribution of $CaCO_3$ dissolution to

$\sum[DIC]_{estuary}$ of ca. 30%, sulfate reduction and aerobic respiration were estimated to contribute 29 to 34% and 36

to 41%, respectively.

### 3.3.3 Evidence from DO

The deficit of $O_2$ in Shark River was found to be $2.7 \pm 0.7 \times 10^6$ and $3.7 \pm 0.3 \times 10^6$ mol during SharkTREx

1 and 2, respectively. Assuming a stoichiometric ratio of ca. 1.1 for $O_2$ to $CO_2$ during degradation/remineralization

of terrestrial organic matter (Severinghaus, 1995; Keeling and Manning, 2014), the maximum contribution of

aerobic respiration to the DIC added to the estuary was estimated to be 57 to 69%. However, $O_2$ may also be

consumed during oxidation of reduced products from anaerobic metabolism, such as $H_2S$, $Mn^{2+}$ or $Fe^{2+}$, with similar

stoichiometry as aerobic respiration. Hence, the numbers derived above represent an upper limit for aerobic

respiration, and if there were complete re-oxidation of metabolites from anaerobic respiration, the $O_2$ deficit would

represent total mineralization of terrestrial organic matter instead of just aerobic respiration. The mangrove

contributions estimated from $\delta^{13}C_{DIC}$ (section 3.3.1) and TAlk/DIC (section 3.3.2) are consistent with this analysis of

the $O_2$ deficit, which indicates that a minimum of 57-69% of $\sum[DIC]_{estuary}$ derived from the mineralization of

organic matter.

## 3.4 Mangrove contributions to DOC inventory

During both experiments, the $\delta^{13}C_{DOC}$ was highly depleted, indicative of contribution from higher plants,

including mangroves. During SharkTREx 1, the lowest observed $\delta^{13}C_{DOC}$ value (-33.8‰) was in the mid-estuary

(i.e., from salinity of ca. 10 to 20) (Fig. 4d), and it was lower than mangrove-derived material, indicative of a highly





reworked organic matter source, and perhaps preferential degradation of enriched compounds had resulted in further
$^{13}$C depletion (Hayes, 1993). The overall $\delta^{13}C_{DOC}$ depletion was less during SharkTREx 2, and the overall
distribution was indicative of a stronger marine influence and/or mixing (Fig. 4e, f). The marine end member had a
more enriched $\delta^{13}C_{DOC}$, perhaps suggesting also a greater contribution of seagrass and/or marine phytoplankton
derived organic matter to the marine DOC pool (Anderson and Fourqurean, 2003). These observations are consistent
with the greater longitudinal dispersion observed during SharkTREx 2 compared to SharkTREx 1.

The calculations of mangrove contribution using $\delta^{13}C_{DOC}$ mass balances (Eq. 10) are in agreement with the
above estimates, and suggest that the majority of $[DOC]_{estuary}$, but only a small percentage of the total DOC
inventory, was derived from mangroves (7 and 5% in the Shark River during SharkTREx 1 and 2, and 7% in the
Harney River during SharkTREx 2).

**3.5    Longitudinal fluxes to the Gulf of Mexico and comparison with previous studies**

Residence times of Shark River (including Tarpon Bay) for SharkTREx 1 and 2 were, 5.8 ± 0.4 and 8.1 ±
1.1 days, respectively (Ho et al., 2016), and that of Harney River was 4.7 ± 0.7 days for SharkTREx 2. The resulting
longitudinal DIC fluxes to the Gulf of Mexico (15.8 to 33.6 x $10^5$ mol d$^{-1}$) were significantly larger than the
longitudinal DOC fluxes (3.3 to 7.5 x $10^5$ mol d$^{-1}$) at salinity of ca. 27 (Fig. 5; Table 2).

There are no previously published DIC inventories or fluxes for the Shark and Harney Rivers, so
comparison with previous studies is focused on the DOC results. The DOC flux from the Shark River to the coastal
ocean in SharkTREx 1 (7.5 ± 0.2 x $10^5$ mol d$^{-1}$) is in very good agreement to that estimated by Bergamaschi et al.
(2011) in an experiment conducted in the Shark River from 20-30 September 2010 (7.6 ± 0.5 x $10^5$ mol d$^{-1}$).
However, the net discharge during the Bergamaschi et al. (2011) study was higher than SharkTREx 1 (9.1 ± 7.1 vs.
6.9 ± 5.3 m$^3$ s$^{-1}$), which would lead to a shorter residence time of 4.6 days using a relationship presented in Ho et al.
(2016). Using the DOC concentration data presented in Bergamaschi et al. (2011) yields an inventory that is ca. 3%
higher than the DOC inventory in Shark River during SharkTREx 1. Calculations using the shorter residence time
and higher DOC inventory yields a DOC flux of 9.7 ± 0.2 x $10^5$ mol d$^{-1}$, which is ca. 30% higher than the estimates
of Bergamaschi et al. (2011). The difference between both methods may be due to the fact that Bergamaschi et al.
(2011) estimated the longitudinal flux at a location ca. 9.5 km from the mouth of the Shark River. Because the DOC
concentration gradients in the middle of the estuary are higher than at the mouth, this may overestimate the flux.




The longitudinal flux of mangrove-derived DOC from Shark River during SharkTREx 1 ($0.3 \pm 0.2 \times 10^5$ mol d$^{-1}$; Table 2) is in rough agreement with the estimate of Cawley et al. (2013) during the same period ($0.2 \times 10^5$ mol d$^{-1}$), but the value for Harney River ($0.6 \pm 0.6 \times 10^5$ mol d$^{-1}$) is lower than their estimate ($1.6 \times 10^5$ mol d$^{-1}$).

Mangroves contributed 4% to 6% of the total longitudinal DOC flux in the Shark River and 7% in the Harney River during SharkTREx 2 (Tables 1 and 4). These calculations are in agreement with those by Cawley et al. (2013), who estimated a mangrove contribution to DOC flux of $3 \pm 10\%$ for Shark River and $21 \pm 8\%$ for the Harney River during November 2010, the same time period as SharkTREx 1. DOC measurements were not made in Harney River as part of SharkTREx 1. However, using the November 2010 DOC data from Harney River collected

by Cawley et al. (2013) for inventory calculations, a mangrove contribution of 19% to the total DOC longitudinal flux to the Gulf of Mexico was obtained.

### 3.6    Distribution of carbon fluxes

During SharkTREx 1 and 2, $\sum[DIC]_{estuary}$ made up 20-28% of the total DIC in the rivers, and $\sum[DOC]_{estuary}$ made up only 4 to 7% of the total DOC in the rivers. Mangroves are estimated to contribute 13 to

19% to the total DIC inventory. In all cases, the mangrove contribution to the DIC inventory is a factor of 3 greater than the mangrove contribution to the DOC inventory (Table 1). During SharkTREx 1 and 2, the inventory of mangrove-derived DIC exceeded that of DOC by a factor of 15 to 17, which supports the idea that a large fraction of the carbon exported by mangroves to surrounding water is as DIC (Bouillon et al., 2008a), but is considerably larger than the estimates of ca. 3 to 10 compiled by Bouillon et al. (2008a) for mangroves at 5 sites in Asia and Africa.

The total dissolved carbon fluxes from all sources (i.e., freshwater wetland, mangrove, carbonate dissolution, and marine input) out of the Shark and Harney Rivers during SharkTREx 1 and 2 are dominated by inorganic carbon (82-83%; see Tables 2 and 4), either via air-water $CO_2$ exchange or longitudinal flux of DIC to the coastal ocean (Fig. 5). The remaining 17-18% of the export is as DOC. This proportioning is remarkably similar between SharkTREx 1 and 2, and between the Shark and Harney Rivers (Table 1). The estuarine contribution to

these fluxes is relatively small (generally <15%), with the exception of air-water $CO_2$ flux, where the estuary contribution was 49 to 63% (Table 4).

In this study, the particulate organic carbon (POC) flux was not examined. However, He et al. (2014) estimated the mangrove-derived POC flux in Shark River by taking the total volume discharge from the five major




rivers along the southwest coast of Everglades National Park from 2004 to 2008, and assuming that Shark River contributed 14% to the mean annual discharge. They then multiplied this discharge by the average POM concentration ($5.20 \pm 0.614$ mg L$^{-1}$) in the middle of the estuary to yield an annual POM flux from Shark River. Based on analysis of organic matter biomarkers, He et al. (2014) estimated that mangrove-derived POM was 70–90% of the total POM pool in the Shark River. Using this contribution and further assuming that 58% of POM weight is POC (Howard, 1965), they estimated a POC flux of 1.0 to 2.2 x 10$^4$ mol d$^{-1}$. Because this estimate was

based on biomarker and POM data from the mid-estuary, where the POM concentration and the mangrove contribution to POM are both likely to be much higher than either toward the freshwater end member or the marine end member, it is likely an overestimate of the mangrove derived POC flux. Nevertheless, the mangrove-derived POC flux determined by He et al. (2014) is still only a small fraction (3 to 7%) of the mangrove-derived dissolved carbon fluxes in Shark River during SharkTREx 1 and 2.

**3.7    Mangrove contributing area and estuary carbon balance**

One of the challenges of relating the results reported here to other studies is to scale the results to a mangrove contributing area, and thereby relate the findings to mangrove forest carbon balance, typically expressed on an aerial basis. Estimates of forest carbon export derived here are compared with other investigations in this estuary. The entire area of mangroves surrounding the Shark and Harney Rivers region is ca. 111 km$^2$, and the water

area is ca. 17.5 km$^2$ (Ho et al., 2014). Scaling the forest area by the water area of Shark River (2.5 km$^2$) yields an associated forest area of 15.9 km$^2$. The forest area associated with Harney River (2.8 km$^2$) is 17.4 km$^2$.

Using the total forest area associated with Shark River to scale estimates of total export of mangrove-derived carbon (the combination of longitudinal fluxes and air-water gas exchange) suggests an average dissolved carbon lateral export rate from the forest of 18.9 to 24.5 mmol m$^{-2}$ d$^{-1}$, including both DIC and DOC. However,

since it is unknown what fraction of the total forest area associated with these rivers exported dissolved carbon through tidal pumping (a function of tidal height and duration), this is considered to be a minimum estimate. Average water levels at high tide during SharkTREx 1 and 2 at the USGS Shark River station were 88% and 95% of maximum wet season water levels reported at this site over the period from November 2007 to December 2012 (U.S. Geological Survey, 2016), and 12 inundation events occurred during both SharkTREx 1 and 2. Water levels in

the main river channel at the USGS Shark River station were above an estimate of the average minimum ground surface elevation derived from nearby groundwater monitoring wells in the estuary (sites SH3 and SH4;





http://sofia.usgs.gov/eden/stationlist.php) for 21% and 28% of the time during the SharkTREx 1 and 2 experimental periods, respectively. These values indicate the export of dissolved carbon from flooded portions of the forest during the discontinuous inundation periods should be significantly greater than the dissolved carbon lateral export rate

derived above in order to produce the observed inventories of mangrove-derived dissolved carbon in the main channel.

Bergamaschi et al. (2011) proposed an annual total DOC export from the forest surrounding Shark River of $15.1 \pm 1.1$ mol m$^{-2}$ y$^{-1}$ and describe their method of calculating contributing area using a model based on the relationship between discharge volume and changes in water levels during tidal cycles. They do not provide a

contributing area, but this can be calculated from their results. They determined longitudinal DOC fluxes of $7.6 \pm 0.5 \times 10^5$ and $1.3 \pm 0.02 \times 10^5$ mol d$^{-1}$ for the wet and dry seasons, respectively, and assumed that they are entirely of mangrove origin. Given the lengths of the wet and dry seasons, this would yield a mean annual DOC flux of $3.9 \pm 0.2 \times 10^5$ mol d$^{-1}$, and $9.4 \pm 0.7$ km$^2$ of mangrove forest contributing to carbon fluxes thru tidal flushing in this segment of Shark River. However, data from SharkTREx 1 and 2 indicate that ca. 5% of the total longitudinal DOC

fluxes were of mangrove origin, with an average mangrove-derived DIC to DOC flux ratio of 10.5. Using this information, the Bergamaschi et al. (2011) results were recalculated to yield a wet season dissolved carbon lateral export rate of $46.5 \pm 4.4$ mmol m$^{-2}$ d$^{-1}$ (as DIC and DOC) from the forest.

Another method of estimating forest lateral carbon export utilizes the difference between measurements of net ecosystem-atmosphere $CO_2$ exchange (NEE) above the mangrove forest surrounding Shark River ($267 \pm 15$

mmol m$^{-2}$ y$^{-1}$ in 2004; (Barr et al., 2012) and corresponding measures of net ecosystem carbon balance (NECB; $227 \pm 14$ mmol m$^{-2}$ d$^{-1}$). NECB in 2004 can be estimated as the sum of carbon in litter fall ($104 \pm 8$ mmol m$^{-2}$ d$^{-1}$), wood production ($44 \pm 3$ mmol m$^{-2}$ d$^{-1}$) (Castañeda-Moya et al., 2013),  root growth ($47 \pm 11$ mmol m$^{-2}$ d$^{-1}$) (Castañeda-Moya et al., 2011) and soil carbon accumulation (31.7 mmol m$^{-2}$ d$^{-1}$) (Breithaupt et al., 2014) measured at the same location (FCE LTER site SRS6) in this forest. The difference between NEE and NECB ($40 \pm 17$ mmol m$^{-2}$ d$^{-1}$)

provides an estimate of the annual rate of forest carbon export to Shark River on a daily basis (Chapin et al., 2006).

The rate of mangrove-derived carbon exported to estuarine waters is likely to vary over space and time, as a result of factors that include tidal cycles, phenology, and forest and soil structural characteristics. For example, Bergamaschi et al. (2011) found that DOC fluxes were 6 times higher during the wet season (September) than the dry season (April), whereas Cawley et al. (2013) found that the DOC fluxes were 4 and 10 times higher during the




wet vs. dry season (November vs. March) in the Shark and Harney Rivers, respectively. Barr et al. (2013) showed that forest respiration rates derived from NEE data are greater during the wet than dry seasons. Higher respiration rates combined with increased inundation during the wet compared to dry seasons suggest that wet season DIC export will also be greater than dry season values. For these reasons, the annual carbon export rates derived from the difference between NECB and NEE are expect to underestimate wet season values. If annual lateral carbon export

rates are considered as equivalent to a time-weighted sum of dry season (7 months) and wet season (5 months) values (after Bergamaschi et al. 2011), and wet season export is assumed to be, for example, 5 times greater than dry season values, the seasonal export rates (15 and 75 mmol $m^{-2}$ $d^{-1}$ for dry and wet seasons, respectively) that correspond with the difference between annual NECB and NEE can be calculated.

The discrepancies between the estimates of carbon export rates derived here, and those derived from

Bergamaschi et al. (2011) and the difference between NEE and NECB point out the need for additional studies to reduce the uncertainty in the relationships between riverine carbon fluxes, forest carbon export, and estimates of contributing areas. For example, Bergamaschi et al. (2011) conducted an Eulerian study at a single location in the middle of the estuary, where the mangrove influence might be higher than the Lagrangian study conducted during SharkTREx 1 and 2, which covered the entire estuary. Also, the estimate of forest carbon export based on the

difference between NEE and NECB is from a single location along Shark River (at FCE LTER site SRS6), and may not be representative of the entire forest. Furthermore, forest lateral carbon export rates and contributing areas should be considered dynamic, varying over semi-diurnal time scales with the extent and duration of inundation during individual tidal cycles. The correct interpretation of a single, static value for contributing area such as derived above is therefore uncertain, since the tracer-based results represent an integration of carbon sources and sinks

calculated over the water residence time and expressed on daily time scales. To improve understanding of how mangrove forest carbon balance and export influence riverine carbon inventories and fluxes to the Gulf of Mexico in this system, wet and dry season measurements over multiple years, information on the relationships between forest structure, productivity and lateral carbon export rates, and independent estimates of forest inundation area in relation to tidal height are needed.

## 4    Conclusions

The SharkTREx 1 and 2 studies are the first to provide estimates of longitudinal DIC export, air-water $CO_2$ fluxes, and mangrove-derived DIC inputs for the Shark and Harney Rivers. The results show that air-water $CO_2$ exchange and longitudinal DIC fluxes account for ca. 90% of the mangrove-derived dissolved carbon export out of the Shark and Harney Rivers, with the remainder being exported as dissolved organic carbon.

The mangrove contribution to the total longitudinal flux was 6.5 to 8.9% for DIC and 4 to 18% for DOC. A lower bound estimate of the dissolved carbon export (DIC and DOC) from the forest surrounding Shark River during the wet season was 18.9 to 24.5 mmol m$^{-2}$ d$^{-1}$ with 15.9 km$^2$ of mangrove contributing area. Other independent estimates of lateral export from this mangrove forest are somewhat higher by comparison. However, mangrove forest carbon export rates on an aerial basis are expected to vary with the spatial and temporal scales over which they are calculated, and depend on factors such as tidal inundation frequency, distance from the riverbank and the coast, and forest and soil characteristics.

Future experiments should investigate the contribution of DIC from groundwater to the rivers, by making measurements of $\delta^{13}C_{DIC}$ of groundwater, Sr and Ca concentrations in the river to quantify $CaCO_3$ dissolution and to separate carbonate alkalinity from TAlk, radon to quantify groundwater discharge, $^{14}C_{DIC}$ to separate input of DIC from remineralization of organic matter from dissolution of $CaCO_3$. Experiments should also examine the seasonal variability in the carbon dynamics and export, by conducting process-based studies like SharkTREx during both wet and dry seasons. Also, time series measurement of current velocities, wind speeds, $pCO_2$ and pH (to calculate DIC), DO, chromophoric dissolved organic matter (CDOM, as a proxy for DOC), and radon will also allow the temporal variability of the sources and sinks of DIC in these rivers to be examined.

## Author contribution

D. Ho, S. Ferron, and V. Engel conceived and executed the experiment, interpreted the data, and prepared the manuscript with input from the other authors. W. Anderson measured the samples for $\delta^{13}C$ of dissolved organic carbon, P. Swart measured the samples for $\delta^{13}C$ of dissolved inorganic carbon, R. Price measured the total alkalinity samples for SharkTREx 1, and L. Barbero measured the total alkalinity and dissolved inorganic carbon samples for SharkTREx 2.

## Data availability

The $pCO_2$ data collected during SharkTREx 1 and 2 are available from the SOCAT database <www.socat.info>.



The other data may be obtained by contacting the corresponding author.

**Acknowledgments**

We thank J. Barr, T. Custer, L. Larsen, M. Reid, and M. Vázquez-Rodriguez for assistance in the field, A. Arik, N. Coffineau and J. Harlay for assistance with data analysis, R. Wanninkhof and R. Zeebe for helpful discussions and comments, M. Sukop for the use of his laboratory, P. Sullivan for analyzing the total alkalinity samples during SharkTREx 1. K. Kotun and Everglades National Park provided boats, fuel, and logistical support for the experiment. Shark River flow velocity data were obtained from USGS via the National Water Information System.

Funding was provided by National Park Service through the Critical Ecosystem Studies Initiative (Cooperative Agreement H5284-08-0029) and by National Science Foundation through the Water Sustainability and Climate solicitation (EAR 1038855). R.M.P. was supported by the Florida Coastal Everglades Long-Term Ecological Research program under National Science Foundation Grant Nos. DBI-0620409 and DEB-1237517. This is SERC contribution number ####. Any use of trade, product, or firm names is for descriptive purposes only and does not

imply endorsement by the U.S. Government.



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





**Table 1.** Inventories of DIC and DOC in Shark and Harney Rivers, as well as contributions from estuarine and non-estuarine sources.

| | | SharkTREx 1 | | | | SharkTREx 2 | | | | | | | |
| | | *Shark River* | | | | *Shark River* | | | | *Harney River* | | | |
| | | Inventory (x10⁶ mol) | Percentage of total Inventory [a] | Contribution to mangrove carbon [b] | Contribution to total carbon [c] | Inventory (x 10⁶ mol) | Percentage of total Inventory [a] | Contribution to mangrove carbon [b] | Contribution to total carbon [c] | Inventory (x 10⁶ mol) | Percentage of total Inventory [a] | Contribution to mangrove carbon [b] | Contribution to total carbon [c] |
|---|---|---|---|---|---|---|---|---|---|---|---|---|---|
| **DIC** | Observed | 19.5 ± 0.9 | - | | | 15.2 ± 1.3 | - | | | 7.4 ± 0.4 | - | | |
| | Gas Exchange | 2.5 ± 0.2 | 11% | | | 3.2 ± 0.1 | 17% | | | 1.5 ± 0.3 | 17% | | |
| | Non-estuarine | 17.6 ± 0.7 | 80% | 94% | 82% | 13.6 ± 1.2 | 74% | 93% | 79% | 6.5 ± 0.4 | 72% | 93% | 79% |
| | Estuarine | 4.4 ± 1.1 | 20% | | | 4.8 ± 1.8 | 26% | | | 2.5 ± 0.6 | 28% | | |
| | Mangrove | 2.9 ± 0.8 | 13% | | | 3.1 ± 1.2 | 17% | | | 1.7 ± 0.4 | 19% | | |
| **DOC** | Observed | 4.4 ± 0.1 | | | | 4.1 ± 0.4 | | | | 1.9 ± 0.1 | | | |
| | Non-estuarine | 4.2 ± 0.1 | 96% | 6% | 18% | 3.9 ± 0.4 | 94% | 7% | 21% | 1.8 ± 0.1 | 93% | 7% | 21% |
| | Estuarine[d] | 0.2 ± 0.1 | 4% | | | 0.2 ± 0.6 | 6% | | | 0.1 ± 0.2 | 7% | | |

[a] The DIC inventory is relative to the total DIC (i.e., $\sum[DIC]_{observed} + \sum[DIC]_{gasex}$).

[b] Contribution of each form of carbon (i.e., DIC, DOC) to the total mangrove-derived carbon pool.

[c] Contribution of each form of carbon (i.e., DIC, DOC) to the total carbon pool.

[d] Estuarine DOC is assumed to be entirely of mangrove origin.



**Table 2.** Longitudinal DIC and DOC fluxes, and air-water $CO_2$ fluxes for the Shark and Harney Rivers during SharkTREx 1 and 2.


| | SharkTREx 1 | | SharkTREx 2 | |
|---|---|---|---|---|
| | **Shark River** | **Harney River** | **Shark River** | **Harney River** |
| *Longitudinal DIC Fluxes (x $10^5$ mol $d^{-1}$)* | | | | |
| Total | 33.6 ± 1.6 | N/A | 18.8 ± 1.6 | 15.8 ± 0.9 |
| Non-estuarine contribution | 30.3 ± 1.1 | N/A | 16.8 ± 1.5 | 13.7 ± 0.8 |
| Estuarine Contribution | 3.3 ± 1.9 | N/A | 2.0 ± 2.2 | 2.1 ± 1.3 |
| Mangrove Contribution | 2.2 ± 1.3 | N/A | 1.3 ± 1.5 | 1.4 ± 0.8 |
| *Air-Water $CO_2$ Fluxes (x $10^5$ mol $d^{-1}$)* | | | | |
| Total | 4.2 ± 0.4 | 4.1 ± 0.2 | 4.0 ± 0.2 | 3.1 ± 0.6 |
| Non-estuarine contribution | 2.1 ± 0.2 | 2.0 ± 0.1 | 1.9 ± 0.1 | 1.1 ± 0.2 |
| Estuarine Contribution | 2.1 ± 0.4 | 2.1 ± 0.3 | 2.1 ± 0.2 | 2.0 ± 0.6 |
| Mangrove Contribution | 1.4 ± 0.3 | 1.4 ± 0.2 | 1.4 ± 0.1 | 1.3 ± 0.4 |
| *Longitudinal DOC Fluxes (x $10^5$ mol $d^{-1}$) [a]* | | | | |
| Total | 7.5 ± 0.2 | 3.3 ± 0.4 | 5.1 ± 0.5 | 4.2 ± 0.2 |
| Non-estuarine contribution | 7.2 ± 0.1 | 2.6 ± 0.3 | 4.8 ± 0.5 | 3.9 ± 0.2 |
| Estuarine Contribution [b] | 0.3 ± 0.2 | 0.6 ± 0.6 | 0.3 ± 0.7 | 0.3 ± 0.3 |

[a] Data for DOC concentration in Harney River during SharkTREx 1 taken from *Cawley et al.* (2013).

[b] Estuarine contribution to DOC is assumed to be entirely of mangrove origin.





**Table 3.** Mangrove contribution to $\sum[\text{DIC}]_{estuary}$ determined from $\delta^{13}C_{DIC}$ mass balance and TAlk/DIC ratios.


| River | Experiment | Methods | |
|---|---|---|---|
| | | $\delta^{13}C_{DIC}$ | TAlk/DIC |
| Shark River | SharkTREx 1 | 60 ± 6% | 70 ± 3% |
| | SharkTREx 2 | 61 ± 6% | 70 ± 3% |
| Harney River | SharkTREx 1 | - | - |
| | SharkTREx 2 | 61 ± 6% | 70 ± 2% |



**Table 4.** Distribution of total and mangrove fluxes of DIC and DOC for Shark and Harney Rivers during SharkTREx 1 and 2.

| | | SharkTREx Experiment # | Estuarine Contribution [a] | Percent of Total Export Flux [b] | Percent of total Mangroves Flux [c] |
|---|---|---|---|---|---|
| Longitudinal DIC Flux | Shark River | 1 | 10% | 74% | 57% |
| | | 2 | 11% | 67% | 45% |
| | Harney River | 1 | - | - | - |
| | | 2 | 13% | 68% | 48% |
| Air-Water $CO_2$ Flux | Shark River | 1 | 49% | 9% | 35% |
| | | 2 | 52% | 14% | 45% |
| | Harney River | 1 | 51% | - | - |
| | | 2 | 63% | 14% | 43% |
| All DIC Fluxes | Shark River | 1 | | 83% | 92% |
| | | 2 | | 82% | 90% |
| | Harney River | 1 | | - | - |
| | | 2 | | 82% | 91% |
| Longitudinal DOC Flux | Shark River | 1 | 4% | 17% | 8% |
| | | 2 | 6% | 18% | 10% |
| | Harney River | 1 | 19% | - | - |
| | | 2 | 7% | 18% | 9% |

[a] Estuarine contribution to the individual fluxes in each river during each experiment

[b] Flux as a percentage of the total dissolved carbon flux (i.e., longitudinal DIC, DOC and air-water $CO_2$ fluxes)

[c] Flux as a percentage of the total mangrove-derived dissolved carbon flux (i.e., longitudinal DIC, DOC and air-water $CO_2$ fluxes)




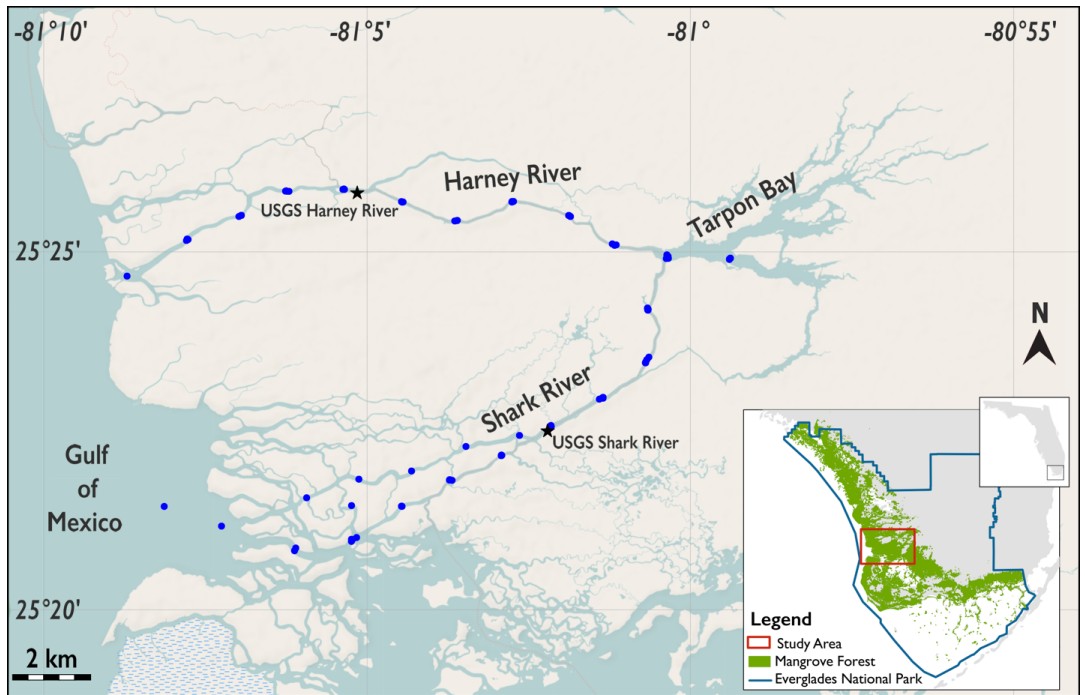

**Figure 1.** Map of the study area near the southern tip of Florida, USA, showing locations of Shark River, Harney River, and Tarpon Bay. The blue circles indicate the locations where discrete samples were taken, and the black stars denote the USGS gaging stations on both rivers. The green areas in the inset are part of the largest contiguous

mangrove forest in North America. Indicated in the inset are the boundaries of Everglades National Park.

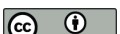



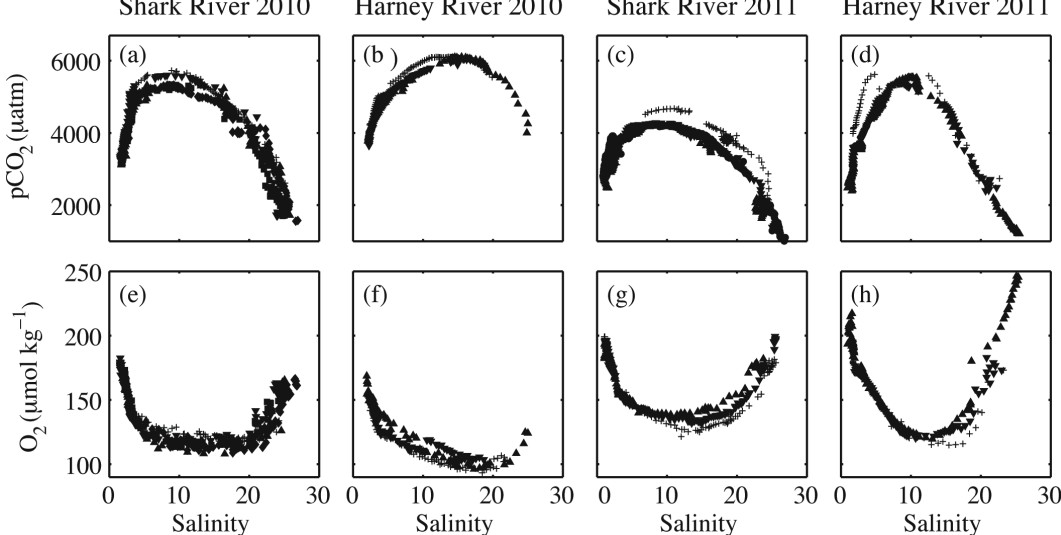

**Figure 2.** Distributions of $pCO_2$ (a-d) and dissolved $O_2$ (e-h) along the salinity gradient in the Shark and Harney Rivers during the 2010 (SharkTREx 1) and 2011 (SharkTREx 2) campaigns. Different symbols represent measurements made on different days.






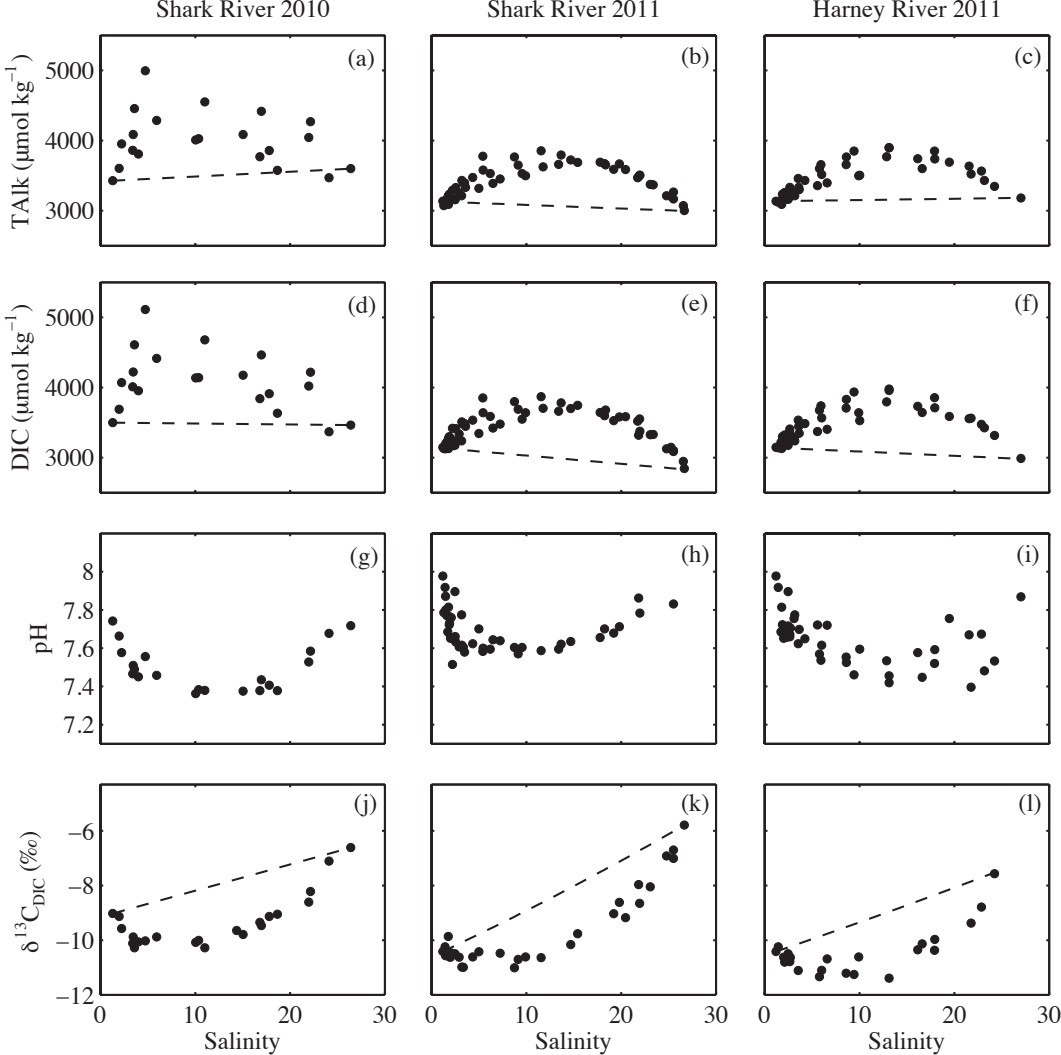

**Figure 3.** Distribution of TAlk (a-c), DIC (d-f), pH (g-i) and $\delta^{13}C_{DIC}$ (j-l) along the salinity gradient in the Shark and Harney Rivers during the 2010 (SharkTREx 1) and 2011(SharkTREx 2) campaigns. During SharkTREx 1, TAlk and pH were measured at FIU, and DIC was calculated using CO2SYS (Pierrot et al., 2006). During SharkTREx 2, DIC and TAlk were measured at NOAA/AOML, and pH was calculated using CO2SYS. The dashed lines indicate the distribution expected for conservative mixing.





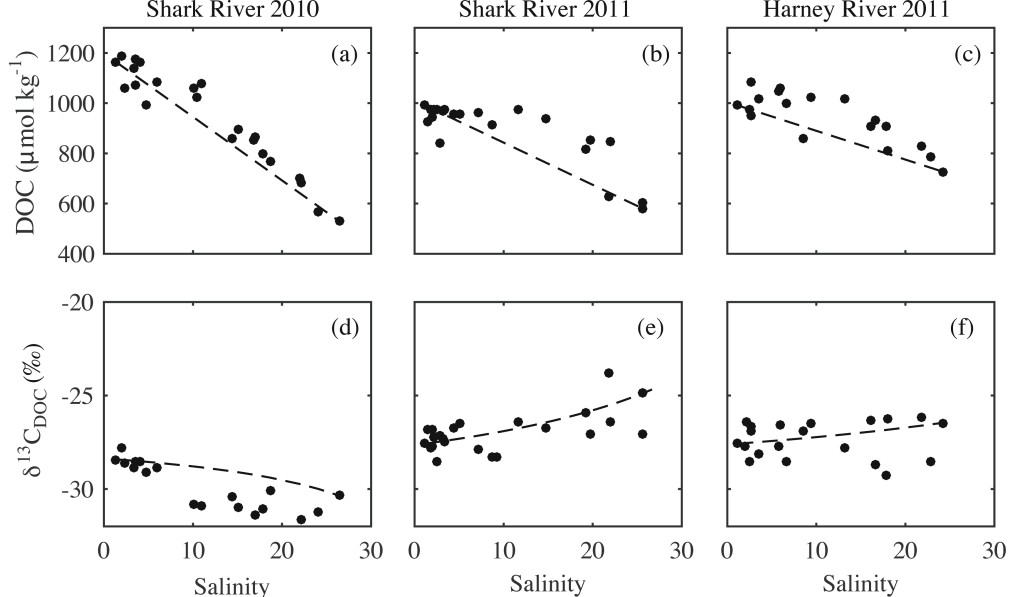

**Figure 4.** Distribution of DOC and $\delta^{13}C_{DOC}$ along the salinity gradient in the Shark and Harney Rivers in samples

collected during SharkTREx 1 and 2. The dashed lines indicate the distribution expected for conservative mixing.



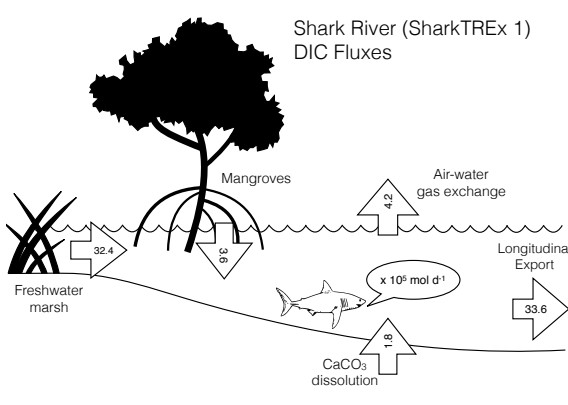

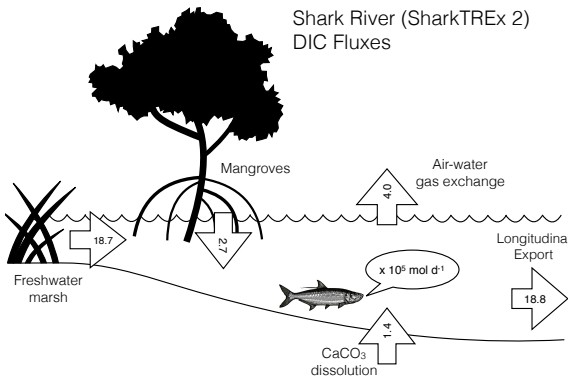

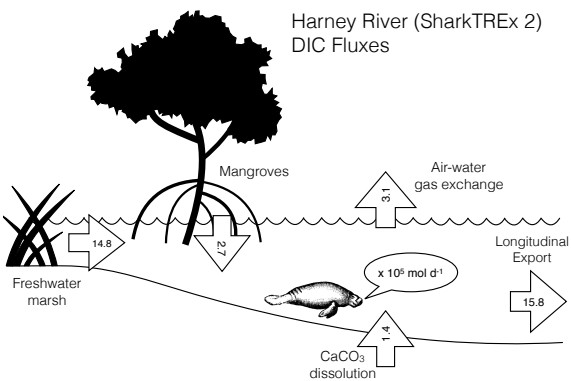

**Figure 5.** Diagrams showing the main DIC fluxes (in $10^5$ mol d$^{-1}$) entering and exiting the Shark and Harney Rivers during SharkTREx 1 and 2. Fluxes from the freshwater marsh were assumed to be fluxes estimated from the conservative DIC curves.





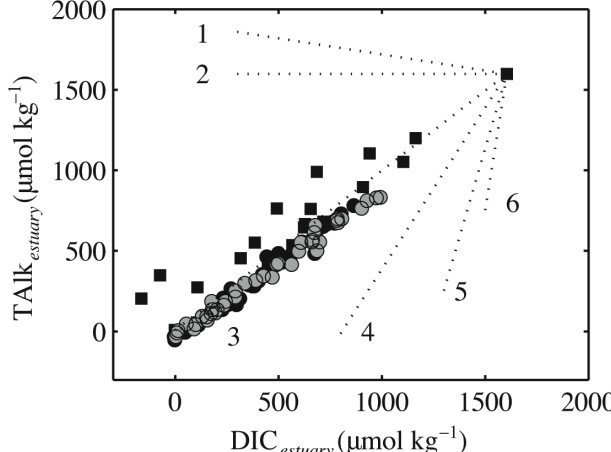

**Figure 6.** (a) Covariation of $DIC_{estuary}$ and $TAlk_{estuary}$. Black squares are samples from the Shark River during SharkTREx 1,
and black and gray circles are from the Shark and Harney Rivers, respectively, during SharkTREx 2. Dotted lines represent
the theoretical covariation of DIC and TAlk for different biogeochemical processes: 1) aerobic respiration; 2) $CO_2$ emission,
3) sulfate reduction, 4) $CaCO_3$ dissolution, 5) manganese reduction, and 6) iron reduction.