# Peer review of "Dissolved carbon biogeochemistry and export in mangrovedominated rivers of the Florida Everglades"

_Biogeosciences, 2017_

## Referee Comment (RC1) · Anonymous Referee #1 · 26 Jan 2017

The authors investigated the dissolved carbon source-sink dynamics in two rivers in order to integrate and estimate the export of carbon from a large mangrove forest, "the largest contiguous mangrove forest in North America". First of all, I must note that it is refreshing to read a manuscript that is written in proper English. I review about one manuscript per week and would put this one in the top 10% in terms of readability. I commend the authors for it. Nevertheless, I did make a number of editorial recommendations in the annotated manuscript that accompanies this review. On the other hand, whereas the paper addresses a relevant scientific question, the paper lacks scientific rigor, the methodologies are only sparsely described, the data processing is questionable if not inadequate and, consequently, the discussion is weak and interpretations

are often speculative given the lack of supporting data. The manuscript is clearly not suitable for publication in its present state. Specific comments: Methods – Line 113: IUPAC has done away with 'equivalents', alkalinities are now reported in moles.

Lines 116, 128, 212: The Millero (2010) constants are not particularly dependable. Perhaps better to use the constants of Cai and Wang (1998). See: Orr, J. C., Epitalon, J. M., and Gattuso, J. P. (2015). Comparison of ten packages that compute ocean carbonate chemistry. Biogeosciences 12(5), 1483–1510.

Lines 127-128: Was TAlk corrected for silicate and phosphate contributions? What about the contributions of organic alkalinity? The latter can be very significant in the case of estuaries draining marshes and organic-rich soils, often accounting for the circum-neutral or even acidic nature of these waters. Uncorrected TAlk values, combined with pCO2 measurements, would generate erroneous DIC values. The authors note that DOC makes up about 20% of the DIC, it surely has a large (negative) contribution to Talk.

Lines 135-136: How were the DOC measurements calibrated and what was their reproducibility and accuracy?

Lines 144-145: What was the reproducibility of the $\delta$13C measurements?

Line 154: What were the $\delta$13C standards?

Lines 158-159: What was the time constant of the probe? These probes have a slow response and do not provide real-time measurements. The quality of the measurements will depend on the cruising speed of the boat or current velocity. Hence, these measurements may carry a spatial (and temporal) uncertainty.

Line 160: The time constant of the optode is even larger than the galvanic sensor. Although the optode response does not drift as much as the galvanic sensor, the precision of the optode is significantly worse than the galvanic sensor. What is the uncertainty on these two measurement methods?

Lines 163-164: Even if the authors refer the reader to detailed descriptions in other papers, they should briefly describe the method and its uncertainties.

Lines 200, 206: Are the gas transfer velocities scaled for wind velocity or turbulence generated by flow in the shallow estuary? The authors should provide the estimated value of the gas transfer velocity in the study setting.

Lines 219-220: The seawater that intrudes at these shallow depths is likely to be supersaturated with respect to calcite and aragonite, but could become undersaturated by accumulating metabolic $CO_2$ generated by microbial degradation of dissolved or particulate organic matter in the sediment/karst.

Line 234: As noted above, dissolution in saline groundwaters can only occur upon the addition metabolic $CO_2$ to the groundwaters from microbial degradation of organic matter. Hence, how would the authors distinguish this contribution of 13C-depleted DIC from that of the mangrove-derived organic matter?

Line 256: Again, what about the metabolic DIC contribution from groundwaters? Mn and Fe oxide reduction will also generate alkalinity.

Line 265, Eqn. (9), 270-271: What about groundwater contributions?

Results and Discussion - Lines 311-312: Somewhat repetitive. Any statistics on this correlation?

Line 326: Very large and surely contributes to the titration alkalinity.

Lines 366-368: I noted this earlier. Why not include it in the mass balance equations?

Line 371: Under what conditions would this occur? Any evidence that aragonite and high Mg-calcites are being dissolved and replaced by low Mg-calcite?

Lines 375, 382: Where and under what conditions is $CaCO_3$ being dissolved? In organic-rich sediments, anaerobic respiration leads to alkalinity production (as indicated earlier), and likely a flux of alkalinity and DIC to the overlying waters. The fluxes

would be modulated by neutralization of alkalinity as it diffuses through the oxic sediment layer and by CaCO3 precipitation in the anoxic sediments.

Line 384: See previous comment. This is speculative in the absence of evidence that this dissolution-precipitation process is active.

Lines 386-388: The authors need to show how these estimates were derived as sulfate reduction does not appear in the mass balance equations.

Lines 394-396: The oxidation of these metabolites does not follow the same stoichiometry as aerobic oxidation of organic matter. In the case of H2S, the redox reaction requires the exchange of 8 electrons (S(-II) to S(+VI), 2 in the case on Mn2+ (to Mn(+IV)) and one for Fe2+ (to Fe(+III)).

Lines 405:-406: This statement makes little sense, unless the preferential degradation of 13C-enriched OM took place before respiration in the mangrove forest.

Please also note the supplement to this comment:
http://www.biogeosciences-discuss.net/bg-2017-6/bg-2017-6-RC1-supplement.pdf

———————————————————————

[Figure]

**Supplement:**

[revised manuscript text omitted]

---

## Referee Comment (RC2) · Anonymous Referee #2 · 6 Mar 2017

Major comments

Mangroves play an important role in carbon cycling on local, regional, and perhaps global scales, and yet there is a lack of detailed budget studies. This paper addresses that need. The suite of measurements—including DIC, DOC, carbon isotopes, oxygen, and alkalinity—is appropriate for the task of determining the sources and sinks of organic and inorganic carbon involving a variety of processes. The measurements show very clear non-conservative behavior in the two tidal rivers studied, strongly suggesting additions of DIC and DOC as water transits through these estuaries.

This paper presents a high-quality suite of measurements from experienced aquatic biogeochemists. The main problem is that the method of creating a carbon budget for

the two estuaries is very confusing. Because of this, it is difficult for me to evaluate the main conclusions of the paper. Therefore, I suggest major revisions.

The technique by which the authors quantify gas fluxes is clear and robust, though error estimates would be helpful. It's the longitudinal fluxes—as the authors call them—that cause confusion. As the authors use the term, the longitudinal flux is averaged over the time period of the tracer release and hence accounts for tidal dispersion as well as the net downstream advective transport. The problem is that the authors use the term too broadly and in ways that defy notions of mass conservation. I specifically take issue with a longitudinal flux due to a certain source, such as freshwater wetlands, mangroves, calcium carbonate, or the ocean (lines 18-21); or due to a certain process, such as estuarine or non-estuarine (lines 289-290 and Table 4), and respiration or dissolution (lines 217-218). A longitudinal flux is due to movement of water and nothing else. It's a term in the mass conservation equation that is separate from internal sources and sinks. The two should not be mixed up.

Also confusing is the alternation between approaches based on inventories and approaches based on fluxes. My suggestion is to focus on a budget approach because this paper is basically about the carbon balance of two estuaries. The final budget is shown in a key figure towards the end of the paper (Figure 5), which clearly shows the control volume approach I am recommending. Remarkably, there is not a single budget equation in the paper. And the method for determining the freshwater wetlands flux shown in this figure should be detailed in the methods, not the figure caption. In the paper, the authors should start with a budget equation and then use inventory equations, such as Equations 2 and 9, as needed to support the budget approach. Budget equations should be shown for DOC, DIC, oxygen, and (perhaps) alkalinity.

A process that appears to be missing in the paper is organic mineralization inside the estuary. The very low DO suggests quite significant net heterotrophy. The authors lump organic mineralization inside the estuary with mangrove root respiration and call it a mangrove source of DIC. Those seem like two very different sources to me, which could

perhaps be distinguished using the isotopes. Figure 5 is remarkable in that it does not show any in situ production or consumption. The net organic matter remineralization should be revealed in the DO budget of the individual terms. The "Evidence from DO" section (3.3.3) should basically detail the DO budget: gas exchange, upstream input, downstream export, net in situ consumption, etc. And this budget could be nicely shown in an additional panel on Figure 5.

Photosynthesis in the river is ignored, which may be reasonable, but the authors need to be more convincing (lines 228-231). Saying chlorophyll is low is not enough. They state that diurnal $pCO_2$ variations are small but provide no data to support that. Please be quantitative.

The choice of endmembers is not clear. How exactly are these chosen and what are the values?

The use of X +/- Y throughout the text (lines 192, 299, 319, 333, 343, 375, etc.) and tables (1, 2, and 3) is unclear. How are X and Y computed? Am I supposed to take Y as a standard deviation or a standard error? What is the sample size for computing X and Y.

Minor comments

Line 25: I think the area referred to here in this flux is the forest area, but the authors should make it clear to remove any ambiguity.

Line 26: To be clear that you are not talking about the Everglades in general, I would replace "in this region" with "for the Shark and Harney Rivers"

I was pleased to see that the authors recognize the likelihood of large seasonality in the carbon cycle of south Florida estuaries (Section 3.7). However, the authors give no indication as to why they chose to sample in November two years in a row instead of sampling once in the wet season and once in the dry season, which would have given them a sense of the importance of seasonality. All I am asking for here is a

few sentences in the intro or methods describing the rationale for the sampling periods chosen.

---

## Author Response (AR1)

Reply to Reviewer 1

Specific comments: Methods – Line 113: IUPAC has done away with 'equivalents', alkalinities are now reported in moles.

We thank the reviewer for catching this. It is now in $\mu$mol kg$^{-1}$.

Lines 116, 128, 212: The Millero (2010) constants are not particularly dependable. Perhaps better to use the constants of Cai and Wang (1998). See: Orr, J. C., Epitalon, J. M., and Gattuso, J. P. (2015). Comparison of ten packages that compute ocean carbonate chemistry. Biogeosciences 12(5), 1483–1510.

We thank the reviewer for point out the interesting discussion about $K_1$ and $K_2$ of Millero (2010) in Orr et al. (2015). Using the constants of Cai and Wang (1998) leads to calculated DIC that ranged from being 0.23% lower, to 0.09% higher than those calculated from Millero (2010), with an average of 0.09% lower. That is, the difference is pretty insignificant. Nevertheless, we have chosen to use Cai and Wang (1998) to calculate DIC.

Lines 127-128: Was TAlk corrected for silicate and phosphate contributions? What about the contributions of organic alkalinity? The latter can be very significant in the case of estuaries draining marshes and organic-rich soils, often accounting for the circum-neutral or even acidic nature of these waters. Uncorrected TAlk values, combined with pCO2 measurements, would generate erroneous DIC values. The authors note that DOC makes up about 20% of the DIC, it surely has a large (negative) contribution to Talk.

TAlk was not corrected for silicate and phosphate. As we state in Lines 127-130: "The measured TAlk and pCO$_2$ from SharkTREx 2 were used to calculate DIC using CO2SYS (Pierrot et al., 2006) and the dissociation constants of (Millero, 2010), and the results were 1.3 ± 1.1% (range: -2.4 to +4.4%) higher than the measured DIC, possibly indicating a slight contribution (ca. 1%) to TAlk from organic or particulate material, as the samples were not filtered."

The fact that the calculated DIC agree well with the measured DIC means that the contribution from organic alkalinity was minimal (ca. 1%), assuming that the dissociations constants used to calculate DIC are correct.

Lines 135-136: How were the DOC measurements calibrated and what was their reproducibility and accuracy?

The TOC analyzer was standardized using 10 and 50 ppm of potassium hydrogen phthalate (KHP), with reagent water as a blank. The analytical precision based on replicates of KHP is ca. ±0.3 ppm. We have added this information to the manuscript.

Lines 144-145: What was the reproducibility of the $\delta^{13}$C measurements?

Reproducibility was 0.2 ‰ as determined by repeated analysis of internal DIC standards.

We have added this information to the manuscript.

Line 154: What were the $\delta^{13}C$ standards?

In order to measure the different isotopic ranges within the collected samples, an isotopic calibration was based on two external standards of potassium hydrogen phthalate (KHP - 29.8‰, OI-Analytical) and glutamine (-11.45‰, Fisher) with a concentration range of 0–25 ppm. These standards were prepared in synthetic seawater to match the sample matrix's salinity. The isotope values of these two standards were determined by using an elemental analyzer isotope ratio mass spectrometer (EA-IRMS). We have added this information to the manuscript

Lines 158-159: What was the time constant of the probe? These probes have a slow response and do not provide real-time measurements. The quality of the measurements will depend on the cruising speed of the boat or current velocity. Hence, these measurements may carry a spatial (and temporal) uncertainty.

See next response.

Line 160: The time constant of the optode is even larger than the galvanic sensor. Although the optode response does not drift as much as the galvanic sensor, the pre-cision of the optode is significantly worse than the galvanic sensor. What is the uncertainty on these two measurement methods?

The stated response time of the galvanic sensor is <10 s to reach 90% of final value, and <16 s to reach 95% of final value. As the reviewer states, the time constant of the optode is longer than the galvanic sensor. It has a stated response time of <25 s to reach 63% of final value. The boat speed during the experiments was typically ca. 3 m/s (ca. 6 knots) (see figure below). The other measurements such as the underway $SF_6$ and $pCO_2$ measurements also have their associated delay between when the water is taken into the $SF_6$ extraction system or $pCO_2$ equilibrator and then the gases finally reach the GC/ECD or NDIR analyzer for analysis. This delay is built-in to the LABVIEW program that assigns GPS position information to the individual measurements based on tests done on the delay in the laboratory. If the delayed response were not corrected, this would lead to a typical offset in location of ca. 50-100 m (less than 1% of the length of the river). Furthermore, most of the measurements are referenced to salinity instead of GPS position, making the exact position less important. Finally, the calculated difference between the effects of conservative mixing and estuarine input on DO in this manuscript were based on relative differences in oxygen, so accuracy of the DO measurements will not significantly affect the results.

[Figure]

Lines 163-164: Even if the authors refer the reader to detailed descriptions in other papers, they should briefly describe the method and its uncertainties.

We have added brief descriptions of the underway $SF_6$ system.

Lines 200, 206: Are the gas transfer velocities scaled for wind velocity or turbulence generated by flow in the shallow estuary? The authors should provide the estimated value of the gas transfer velocity in the study setting.

The gas transfer velocities are affected by both wind and currents, and we have added that to the manuscript. On Lines 191-193, we stated: "$k(600)$ for SharkTREx 1 and 2, determined from the parameterization proposed in Ho et al. (2016), were $3.5 \pm 1.0$ and $4.2 \pm 1.8$ cm h$^{-1}$, respectively." The details for gas exchange in Shark River is given in Ho, D. T., N. Coffineau, B. Hickman, N. Chow, T. Koffman, and P. Schlosser (2016), Influence of current velocity and wind speed on air-water gas exchange in a mangrove estuary, *Geophy. Res. Lett.*, *43*, doi:10.1002/2016GL068727.

Lines 219-220: The seawater that intrudes at these shallow depths is likely to be supersaturated with respect to calcite and aragonite, but could become undersaturated by accumulating metabolic $CO_2$ generated by microbial degradation of dissolved or particulate organic matter in the sediment/karst.

Agreed. We now specify this in the text:

In the manuscript (lines 219-220), we wrote: "Groundwater in this region is likely to contain DIC from $CaCO_3$ dissolution that occurs when saltwater intrudes into the karst aquifer that underlies this region (Price et al., 2006)." We expanded on this with: "Groundwater in this region is likely to contain DIC from $CaCO_3$ dissolution that occurs when saltwater intrudes into the karst aquifer that underlies this region (Price et al., 2006), as well as DIC from sediment organic matter mineralization."

Line 234: As noted above, dissolution in saline groundwaters can only occur upon the addition metabolic $CO_2$ to the groundwaters from microbial degradation of organic matter. Hence, how would the authors distinguish this contribution of 13C-depleted DIC from that of the mangrove-derived organic matter?

We assumed that all organic matter added in the estuary is from the mangroves.

Line 256: Again, what about the metabolic DIC contribution from groundwaters? Mn and Fe oxide reduction will also generate alkalinity.

The DIC from groundwater is assumed to be either from calcite dissolution or from microbial degradation of organic matter. As we specify in the text, the calculation of $[DIC]_{dissolution}$ from $[TAlk]_{estuary}$ provides an upper-bound estimate of the contribution of dissolution to DIC as it does not take into account the contribution of other mineralization pathways to total alkalinity.

In mangrove sediments, aerobic respiration and sulfate reduction are generally the main organic matter degradation pathways, and in our case this is supported by Figure 6, which shows that the contributions of Fe and Mn reduction are likely small.

Also, Fe concentrations in Shark Slough sediments are very low (1.1 mg gdw$^{-1}$; Chambers, R. M., and Pederson, K. A.: Variation in soil phosphorus, sulfur, and iron pools among south Florida wetlands, Hydrobiologia, 569, 63-70, 10.1007/s10750-006-0122-3, 2006), and Mn is expected to be similarly low in this carbonate/mangrove peat setting.

Furthermore, the agreement between measured and calculated DIC (discussed above) indicates that in this environment, the other sources of alkalinity are small, compared to carbonate alkalinity.

Line 265, Eqn. (9), 270-271: What about groundwater contributions?

Groundwater added in the estuary is part of $[DOC]_{estuary}$

Results and Discussion - Lines 311-312: Somewhat repetitive. Any statistics on this correlation?

We have removed this repetitive statement.

Line 326: Very large and surely contributes to the titration alkalinity.

See reply above regarding measured DIC vs. DIC calculated from TAlk and $pCO_2$.

Lines 366-368: I noted this earlier. Why not include it in the mass balance equations?

Because that is either considered to be calcite dissolution or microbial degradation of mangrove organic matter.

Line 371: Under what conditions would this occur? Any evidence that aragonite and high Mg-calcites are being dissolved and replaced by low Mg-calcite?

Since we do not know whether isotopic exchange happens during dissolution and re-precipitation, we have removed this paragraph.

Lines 375, 382: Where and under what conditions is $CaCO_3$ being dissolved? In organic-rich sediments, anaerobic respiration leads to alkalinity production (as indicated earlier), and likely a flux of alkalinity and DIC to the overlying waters. The fluxes would be modulated by neutralization of alkalinity as it diffuses through the oxic sediment layer and by CaCO3 precipitation in the anoxic sediments.

Dissolution/re-precipitation of $CaCO_3$ is omnipresent in Florida, but this was not one of the processes examined during SharkTREx 1 and 2. Previous studies (e.g., Stalker et al, 2009) have shown that, surface waters in Florida show higher Sr/Ca ratio than expected from seawater. Even though Sr/Ca released by dissolution is the same as seawater because aragonite Sr/Ca is the same as seawater Sr/Ca, re-precipitation of calcite rejects Sr, which elevates the Sr/Ca ratio.

Stalker, J. C., R. M. Price, and P. K. Swart: Determining Spatial and Temporal Inputs of Freshwater, Including Submarine Groundwater Discharge, to a Subtropical Estuary Using Geochemical Tracers, Biscayne Bay, South Florida, *Estuar Coast*, *32*, 694-708, doi:10.1007/s12237-009-9155-y, 2009.

Line 384: See previous comment. This is speculative in the absence of evidence that this dissolution-precipitation process is active.

We agree, and have removed this speculative sentence.

Lines 386-388: The authors need to show how these estimates were derived as sulfate reduction does not appear in the mass balance equations.

The ratio of TAlk to DIC for calcite dissolution is 2, and that of sulfate reduction and aerobic respiration are 0.99 and -0.2, respectively. Hence, to achieve the observed ratios of TAlk to DIC of 0.84, 0.92, and 0.90 for the three cases, and given the contribution of calcite dissolution of 30%, the contribution of sulfate reduction and aerobic respiration could be calculated using straightforward algebra (assuming no contribution from Fe and Mn reduction). We have stated this more explicitly in the manuscript.

Lines 394-396: The oxidation of these metabolites does not follow the same stoichiometry as aerobic oxidation of organic matter. In the case of $H_2S$, the redox reaction requires the exchange of 8 electrons (S(-II) to S(+VI), 2 in the case on Mn2+ (to Mn(+IV)) and one for Fe2+ (to Fe(+III)).

Here, we meant $O_2$ to $CO_2$ stoichiometry. Oxygen uptake due to the re-oxidation of reduced metabolites from sulfate, iron, and manganese reduction results in carbon to oxygen stoichiometry that is similar to aerobic respiration. For that reason, the uptake of

oxygen is equivalent to total carbon mineralization if there is complete re-oxidation of metabolites and denitrification is negligible (see Canfield, 1993; Hulth et al. 1999; Reimers et al., 1992). Therefore, the estimate can be considered as a lower-bound estimate of total carbon mineralization.

Canfield, D. E.: Organic Matter Oxidation in Marine Sediments, in: Interactions of C, N, P and S Biogeochemical Cycles and Global Change, edited by: Wollast, R., Mackenzie, F. T., and Chou, L., Springer Berlin Heidelberg, Berlin, Heidelberg, 333-363, 1993.

Hulth, S., Aller, R. C., and Gilbert, F.: Coupled anoxic nitrification/manganese reduction in marine sediments, Geochim. Cosmochim. Acta, 63, 49-66, 10.1016/S0016-7037(98)00285-3, 1999.

Reimers, C. E., Jahnke, R. A., and McCorkle, D. C.: Carbon fluxes and burial rates over the continental slope and rise off central California with implications for the global carbon cycle, Glob. Biogeochem. Cycle, 6, 199-224, 10.1029/92GB00105, 1992.

Lines 405:-406: This statement makes little sense, unless the preferential degradation of 13C-enriched OM took place before respiration in the mangrove forest.

We thank the reviewer for catching this, and have revised the statement. This statement was a remnant from an early draft. Initial $\delta^{13}C_{DOC}$ data did include salinity correction following the method detailed in Ya et al. (2015), where the standards were match to the salinity range of the samples with synthetic seawater mixtures, with a maxim salinity varying from 30 to 32 depending on the timing and reported salinities of the sampling period. In the revised $\delta^{13}C_{DOC}$ data, the lowest observed values from SharkTREx 1 was -31.6 ± 1.25 ‰, as shown in Figure 4d. Previous studies of DOC from mangrove-dominated systems have reported values as low as -30.4‰ (Dittmar, et al, 2006), and some of the more depleted samples from SharkTREx 1 might have DOC sourced from algae associated with mangrove roots, which can have relatively depleted values (Kieckbush et al, 2004).

Dittmar, T. , Hertkorn, N. , Kattner, G. and Lara, R. J.: Mangroves, a major source of dissolved organic carbon to the oceans , *Global Biogeochemical Cycles*, 20, GB1012. doi: 10.1029/2005GB002570, 2006.

Kieckbusch, D.K., Koch, M.S,. Serafy, J.E., and Anderson, W.T.: Trophic linkages of primary producers and consumers in fringing mangroves of tropical lagoons, *Bulletin of Marine Science*, v. 74, no. 2, p. 271-285, 2004.

Ya, C., Anderson, W., and Jaffé, R.: Assessing dissolved organic matter dynamics and source strengths in a subtropical estuary: Application of stable carbon isotopes and optical properties, *Continental Shelf Research*, 92, 98-107, 10.1016/j.csr.2014.10.005, 2015.

Reply to Reviewer 2

Major comments

The technique by which the authors quantify gas fluxes is clear and robust, though error estimates would be helpful. It's the longitudinal fluxes, as the authors call them, that cause confusion. As the authors use the term, the longitudinal flux is averaged over the time period of the tracer release and hence accounts for tidal dispersion as well as the net downstream advective transport. The problem is that the authors use the term too broadly and in ways that defy notions of mass conservation. I specifically take issue with a longitudinal flux due to a certain source, such as freshwater wetlands, mangroves, calcium carbonate, or the ocean (lines 18-21); or due to a certain process, such as estuarine or non-estuarine (lines 289-290 and Table 4), and respiration or dissolution (lines 217-218). A longitudinal flux is due to movement of water and nothing else. It's a term in the mass conservation equation that is separate from internal sources and sinks. The two should not be mixed up.

It is true that longitudinal water flux would be due solely to movement of water. However, longitudinal carbon flux is due to movement of water and the concentration of dissolved carbon in the water. When we talk about contributions, we are not talking about the contribution of the freshwater marsh, mangroves, or carbonate to the water movement, but to the dissolved carbon concentration.

We have clarified this in the revised manuscript. For example, we have changed this sentence in the abstract:

"Approximately 80% of the total dissolved carbon flux from all sources (i.e., freshwater wetlands, mangrove, carbonate dissolution, and marine input) out of the Shark and Harney Rivers during these experiments was as inorganic carbon, either via air-water $CO_2$ exchange or longitudinal flux of inorganic carbon to the coastal ocean."

To:

"Approximately 80% of the total dissolved carbon flux out of the Shark and Harney Rivers during these experiments was in the form of inorganic carbon, either via air-water $CO_2$ exchange or longitudinal flux of dissolved inorganic carbon (DIC) to the coastal ocean.."

In the methods section, we have changed:

"As with the inventory calculations, longitudinal fluxes were separated into estuarine and non-estuarine contributions."

To:

"In addition, using the estuarine and non-estuarine fractions of the inventories in equation (12) allowed the estuarine and non-estuarine proportions of the longitudinal carbon fluxes to be quantified."

Also confusing is the alternation between approaches based on inventories and approaches based

on fluxes. My suggestion is to focus on a budget approach because this paper is basically about the carbon balance of two estuaries. The final budget is shown in a key figure towards the end of the paper (Figure 5), which clearly shows the control volume approach I am recommending. Remarkably, there is not a single budget equation in the paper. And the method for determining the freshwater wetlands flux shown in this figure should be detailed in the methods, not the figure caption. In the paper, the authors should start with a budget equation and then use inventory equations, such as Equations 2 and 9, as needed to support the budget approach. Budget equations should be shown for DOC, DIC, oxygen, and (perhaps) alkalinity.

We recognize that what the reviewer is suggesting is one valid approach, but we have taken what we feel is an equally valid approach. Determining the carbon balance in the estuaries was not our main goal, although in determining the main sources and sinks, we do eventually arrive there.

Our goal is to determine the export (fluxes to the coastal ocean and across the air-water interface) of mangrove derived carbon from the estuary. We took the approach: Flux = Inventory/Residence Time, where residence times were determined from the tracer release experiments. Hence, in order to determine mangrove derived DIC and DOC fluxes, we needed to find out how much DIC and DOC were in the rivers (i.e., the inventory), and also the sources of this carbon (mangrove vs. calcite dissolution). Then, we needed to determine the fate of the mangrove-derived carbon in the river (i.e., lost to the atmosphere vs. lost to the coastal ocean).

We are not alternating between an inventory and flux approach. We are systematically using the inventory to determine the flux.

A process that appears to be missing in the paper is organic mineralization inside the estuary. The very low DO suggests quite significant net heterotrophy. The authors lump organic mineralization inside the estuary with mangrove root respiration and call it a mangrove source of DIC. Those seem like two very different sources to me, which could perhaps be distinguished using the isotopes. Figure 5 is remarkable in that it does not show any in situ production or consumption. The net organic matter remineralization should be revealed in the DO budget of the individual terms. The "Evidence from DO" section (3.3.3) should basically detail the DO budget: gas exchange, upstream input, downstream export, net in situ consumption, etc. And this budget could be nicely shown in an additional panel on Figure 5.

We consider all organic matter added in the estuary as mangrove derived, so the reviewer is correct that we consider DIC produced from organic mineralization inside the estuary as mangrove derived carbon. In the final budget, it does not matter whether the mineralization occurred in the mangrove sediments, or in the river.

Photosynthesis in the river is ignored, which may be reasonable, but the authors need to be more convincing (lines 228-231). Saying chlorophyll is low is not enough. They state that diurnal $pCO_2$ variations are small but provide no data to support that. Please be quantitative.

As we state in the manuscript, photosynthesis in the river is ignored because of low chl *a* and low phytoplankton biomass. With respect to day/night $pCO_2$ variations, we make continuous hourly measurements of $pCO_2$ from Shark River. The average difference in $pCO_2$ during the night (6p to 5:59a local time) is 3% lower than during the day (6a to 5:59p local time), a small

amount and not in the direction one might expect if photosynthesis were significant. We have added this quantification to the manuscript.

The choice of endmembers is not clear. How exactly are these chosen and what are the values?

We write in the manuscript that "the freshwater and marine end-members were assigned to the values measured at the lowest (Tarpon Bay) and highest salinities, respectively." Basically, we went as far as up river and out to the Gulf of Mexico as we could with the boat that we had, and we chose those measurements as the end-members. As we state in the paper, "During SharkTREx 1, the salinity along the longitudinal transects ranged from 1.2 to 27.1, and the mean (± s.d.) water temperature was 23.4 ± 0.2 °C. During SharkTREx 2, salinity ranged from 0.6 to 27.1, and water temperatures averaged 22.7 ± 0.9 °C."

The use of X +/- Y throughout the text (lines 192, 299, 319, 333, 343, 375, etc.) and tables (1, 2, and 3) is unclear. How are X and Y computed? Am I supposed to take Y as a standard deviation or a standard error? What is the sample size for computing X and Y.

Depending on the parameter, the error estimates are based on standard deviations of the measured parameters, or the propagated errors of variables used to calculate the parameter.

Some of the values that the reviewer questioned are quoted from other published papers (e.g., line 192), whereas we do specify that we are quoting the mean and standard deviation in others (e.g., line 299):

"During SharkTREx 1, the salinity along the longitudinal transects ranged from 1.2 to 27.1, and the mean (± s.d.) water temperature was 23.4 ± 0.2 °C. During SharkTREx 2, salinity ranged from 0.6 to 27.1, and water temperatures averaged 22.7 ± 0.9 °C."

We have indicated the number of samples used to calculate mean ± s.d. in the appropriate places in the manuscript, and have specified in the caption to the tables how ± are calculated.

Minor comments

Line 25: I think the area referred to here in this flux is the forest area, but the authors should make it clear to remove any ambiguity.

Yes, we have added language that indicates that flux is from the forest.

Line 26: To be clear that you are not talking about the Everglades in general, I would replace "in this region" with "for the Shark and Harney Rivers"

Done

I was pleased to see that the authors recognize the likelihood of large seasonality in the carbon cycle of south Florida estuaries (Section 3.7). However, the authors give no indication as to why they chose to sample in November two years in a row instead of sampling once in the wet season and once in the dry season, which would have given them a sense of the importance of

seasonality. All I am asking for here is a few sentences in the intro or methods describing the rationale for the sampling periods chosen.

There was no good scientific reason for choosing to conduct the experiments in November, twice. SharkTREx 1 and 2 were piggy-backed on funded experiments that took the team from Hawaii to Florida, and these funded experiments had to be conducted in November. SharkTREx 1 was a pilot experiment, and focused on just the Shark River, had fewer stations, and where some important measurements were not made (e.g., DIC). SharkTREx 2 was a follow up to fill some of the gaps (i.e., more stations, Harney River, and high quality DIC measurements). Future studies will be conducted in the dry season.

[revised manuscript text omitted]

---

## Author Response (AR2)

Response to the editor:

Lines 179 to 181: Was gas exchange accounted for as well and subtracted? If yes, please state.

Yes, we now say that:

"The inventories of DIC and DOC were separated into contributions from estuarine and non-estuarine sources, first by determining inventories for DIC assuming conservative mixing between the freshwater and marine end members and then subtracting these inventories from the total observed inventories while correcting for air-water gas exchange."

The gas exchange component is in fact added, because DIC is lost due to gas exchange in the estuary.

We do state this further on, so the editor might be pointing out that we're being a bit repetitive:

The estuarine DIC inventory, $\sum[DIC]_{estuary}$, representing the DIC from all estuarine sources, was calculated as follows:
$$\sum[DIC]_{estuary} = \sum[DIC]_{observed} - \sum[DIC]_{conserv} + \sum[DIC]_{gasex}, \qquad (2)$$
where $\sum[DIC]_{conserv}$ is the inventory of DIC assuming conservative mixing between freshwater and marine end members (i.e., from non-estuarine sources), and $\sum[DIC]_{gasex}$ is the inventory of DIC lost to air-water gas exchange from the estuary, due to $pCO_2$ in the water being above solubility equilibrium with the atmosphere (see section 2.6).

Line 250: Should this read "Equation 6 and 7" rather than "Equations 6 and 8"?

We really did mean Equations 6 and 8, but the statement is confusing and perhaps awkward. We've changed it to read: "Equation 6 and the following:"

If a reference is available for equation 12, please add.

[revised manuscript text omitted]